# TENT2, TUT4, and TUT7 selectively regulate miRNA sequence and abundance

Acong Yang [1,3], Xavier Bofill-De Ros [1,3], Ryan Stanton [1], Tie-Juan Shao[1,2], Patricia Villanueva[1] & Shuo Gu [1] ✉

TENTs generate miRNA isoforms by 3' tailing. However, little is known about how tailing regulates miRNA function. Here, we generate isogenic HEK293T cell lines in which TENT2, TUT4 and TUT7 are knocked out individually or in combination. Together with rescue experiments, we characterize TENT-specific effects by deep sequencing, Northern blot and in vitro assays. We find that 3' tailing is not random but highly specific. In addition to its known adenylation, TENT2 contributes to guanylation and uridylation on mature miRNAs. TUT4 uridylates most miRNAs whereas TUT7 is dispensable. Removing adenylation has a marginal impact on miRNA levels. By contrast, abolishing uridylation leads to dysregulation of a set of miRNAs. Besides let-7, miR-181b and miR-222 are negatively regulated by TUT4/7 via distinct mechanisms while the miR-888 cluster is upregulated specifically by TUT7. Our results uncover the selective actions of TENTs in generating 3' isomiRs and pave the way to investigate their functions.

MicroRNAs (miRNAs) are a class of regulatory noncoding RNAs. Despite their small size (~22 nt), they play critical roles in development and disease by regulating most, if not all protein-coding genes in mammals[1]. This highlights the importance of deciphering the regulation of miRNAs themselves.

After being transcribed from the genome, miRNAs are embedded in hairpin structures within primary transcripts known as pri-miRNAs. In many cases, multiple miRNAs are clustered within the same pri-miRNA, leading to coupled biogenesis. The miRNA-embedded hairpin is cropped from the pri-miRNA by the ribonuclease Drosha in the nucleus and then translocated to the cytoplasm. This hairpin, termed precursor miRNA (pre-miRNA), is further cleaved by Dicer, resulting in a small RNA duplex[2]. Depending on the end-thermostability, either the 5p- or the 3p-arm of the pre-miRNA joins Argonaute proteins (AGO) to form an RNA-induced silencing complex (RISC), while the other miRNA strand is subsequently degraded[3]. Accordingly, mature miRNAs are designated as 5p- or 3p miRNAs, depending on which strand is used. Once loaded into RISC, miRNAs recognize target mRNAs by sequence complementarity and repress their expression by mRNA degradation and/or inhibiting translation[4].

Terminal nucleotidyltransferases (TENTs) modulate miRNA sequences by adding non-templated (NT) nucleotides to the 3' end, a process referred to as tailing[5–7]. As a result, miRNA isoforms with heterogeneous 3'-ends known as 3' isomiRs have been abundantly detected by deep sequencing[8,9]. Among 11 TENTs identified in mammals, TUT4 and TUT7 are implicated in uridylation of both precursor and mature miRNAs, whereas TENT2 is involved in adenylation[10,11]. While isomiRs with NT tails are clearly a result of TENT-mediated modifications, the origin of isomiRs with "templated tails" is ambiguous. Some of these tails are added by TENTs in an NT manner but nonetheless happen to match the genomic sequence. Others arise from alternative choices of cleavage sites by Drosha and Dicer during biogenesis[12]. The profile of 3' isomiRs is usually cell-type and tissue-specific and can be used as a biomarker to identify many diseases, including cancer[13–15], suggesting that isomiRs have a functional role. Indeed, case studies indicate that isomiRs can have distinct activities in a wide range of biological processes[16–18]. Recently, we demonstrated that 3' uridylation can alter the way miRNAs recognize targets[19], providing one mechanism by which 3' isomiRs possess unique functions. Despite their importance, little is

[1]RNA Mediated Gene Regulation Section; RNA Biology Laboratory, Center for Cancer Research, National Cancer Institute, Frederick, MD 21702, USA. [2]School of Basic Medicine, Zhejiang Chinese Medical University, Hangzhou 310053, China. [3]These authors contributed equally: Acong Yang, Xavier Bofill-De Ros. ✉e-mail: shuo.gu@nih.gov

known about how 3′ isomiRs are generated. It remains unclear whether different TENTs specifically modify various miRNA sequences. This becomes a major challenge in studying the biological function of 3′ isomiRs.

TENTs are known to regulate miRNA abundance via diverse mechanisms. TUT4 and TUT7 negatively regulate let-7 family members by oligo-uridylating their precursors in a LIN28-dependent manner[20–22]. The oligo-uridylated precursors are subsequently degraded by nuclease DIS3L2, leading to lower levels of mature let-7[23,24]. On the other hand, TUT4, TUT7, and TENT2 positively regulate certain let-7 miRNAs by adding a single nucleotide to their pre-miRNAs in the absence of LIN28, promoting Dicer processing, which in turn leads to higher levels of mature miRNAs[25]. In *Drosophila*, uridylation negatively regulates the biogenesis of mirtrons[26,27], a class of Drosha-independent miRNAs generated from introns. In addition, TENTs can modulate miRNA turnover by tailing the mature miRNA. In plants, uridylated miRNAs are removed by 3′ to 5′ exonucleases[28,29]. In mammals, TENT2-mediated adenylation stabilizes certain miRNAs in a cell context-dependent manner[30,31] whereas adenylation of miR-21 by TENT4B leads to its decay[32]. In HEK293T cells, a subset of miRNAs including miR-222 is oligo-uridylated by TUT4 and TUT7 and subsequently degraded by DIS3L2[33]. Finally, TUT4 and TUT7 regulate miRNA abundance by affecting the choice of strand selection. Uridylation of pre-miR-324 promotes the incorporation of its 3p strand over the 5p strand into downstream RISC in glioblastoma cells[34]. It remains to be examined whether additional miRNAs are regulated by TENTs.

Previous efforts studied adenylation and uridylation separately. TENT2, TUT4, and TUT7 and their homologs have been either knocked down or knocked out in various systems including human cells, mouse, and *C. elegans*[35–38]. miRNAs were sequenced accordingly. However, different sequencing approaches were used, making it challenging to compare these datasets directly. Furthermore, rescue assays were missing in most of these studies, making it difficult to parse out the TENT-specific effects. Here, we systematically investigated how TENTs regulate miRNA sequence and abundance by generating a set of human HEK293T cell lines in which TENT2, TUT4, and TUT7 were knocked out individually or in combination. Deep sequencing analyses confirmed that uridylation or adenylation of miRNAs decreased accordingly, and could be rescued by ectopic expression of the corresponding tailing enzyme(s). Surprisingly, we found that TENT2 contributes to guanylation and uridylation in addition to adenylation on mature miRNAs in cells. Despite having comparable in vitro activity, TUT4 and TUT7 function differently in cells. TUT4 uridylates most miRNAs but nonetheless prefers miRNA substrates ending with guanosine. Uridylation mediated by TUT7, on the other hand, is apparently limited to the tailing of pre-miRNAs. While TENT2 specifically modulates miRNA sequences, removing TENT2 has only a marginal impact on miRNA levels in HEK293T cells. By contrast, abolishing uridylation led to specific up- and downregulation of a set of miRNAs. In addition to let-7, we identified miR-181b-5p, miR-222-3p, and miR-888 cluster miRNAs that are regulated by TUT4/7-mediated uridylation via distinct mechanisms. Our results uncover the molecular basis for selective yet coordinated actions of TENTs, highlight the precise control of different 3′ miRNA modifications in cells, and pave the way to investigate their functions.

## Results

### miRNA 3′ tailing is frequent and specific

To measure 3′ tailing on mature miRNAs, we isolated endogenous AGO2 by immunoprecipitation (IP) in HEK293T cells and deep sequenced small RNAs (sRNAs) in the pull-down. More than 7000 3′-isomiRs resulting from 683 miRNAs were consistently detected in two biological replicates (Supplementary Fig. 1a). We focused on isomiRs containing NT nucleotide(s) at the 3′-end, which can only be

generated by tailing. For each miRNA, we measured the percentage of trimmed and/or tailed by the relative abundance of the corresponding isomiRs. On average, more than 18% of miRNA reads had an NT tail (Fig. 1a), arguing that miRNA 3′ tailing is not a minor event. As expected, NT tailing on 3p miRNAs was slightly more prevalent than that on 5p miRNAs (Fig. 1a), attributed to tailings inherited from pre-miRNAs, which only affect 3p miRNAs (Supplementary Fig. 1b). Nonetheless, a significant portion of tailing was observed on 5p miRNAs, indicating that 3′ tailing frequently occurs at mature miRNAs.

NT nucleotides added to the 3′-end of miRNA were not random but highly consistent between two biological replicates (Supplementary Fig. 1c). They were mainly U and A, followed by a small portion (~10%) as G and <1% as C (Fig. 1b). Most NT tails were short, with nearly half consisting of a single nucleotide. While mono-U and mono-A were the most common NT tails, oligo-U and oligo-A tails were rare. Most oligo-tails (length >1 nt) were composed of mixed nucleotides (Fig. 1c). The frequency of 3′ NT tailing varied drastically among different miRNAs. While NT-tailed isomiRs were barely detectable for some miRNAs, they were the dominant form for others (Fig. 1d). For example, only 0.5% of miR-16-5p reads had an NT tail at the 3′-end, whereas more than half of the miR-345-5p reads were NT-tailed (Fig. 1e), indicating that tailing is miRNA-specific. Together, these results suggest that miRNA 3′ tailing is regulated in cells.

To ensure that our analyses were not limited to AGO2-associated miRNAs, we sequenced sRNAs from HEK293T cells total RNAs and compared the result to that from the AGO2-IP. While miRNAs were enriched in AGO2-associated sRNAs as expected (Supplementary Fig. 1d), the relative abundance of isomiRs did not change (Fig. 1f). Examination of top 100 expressed mono-U-tailed and mono-A-tailed isomiRs showed no consistent alteration between total miRNAs and miRNAs associated with the AGO2 (Supplementary Fig. 1e). Furthermore, we sequenced AGO1-associated miRNAs by IP of endogenous AGO1. No preference of uridylation or adenylation was observed between AGO1- and AGO2-associated miRNAs (Supplementary Fig. 1f). To ensure this is not due to a lack of specificity between anti-AGO1 and anti-AGO2 antibodies, we ectopically expressed Flag-tagged AGO1 and AGO2 in HEK293T cells separately and isolated associated sRNAs by FLAG-IP. Again, the isomiR profiles are comparable between AGO1- and AGO2-associated miRNAs (Fig. 1g). These results indicate that the miRNA 3′ modification machinery targets miRNAs associated with different AGOs analogously. Consistent with previous studies[39,40], the overall miRNA expression profiles are also similar (Supplementary Fig. 1f), supporting the notion that miRNAs are not specifically sorted into different AGOs in mammalian cells. Interestingly, uridylated isomiRs were depleted in miRNAs associated with FLAG-AGOs compared to miRNAs associated with endogenous AGOs. This is consistent between AGO1 and AGO2 while being specific to uridylated but not adenylated isomiRs (Supplementary Fig. 1g). Given that newly produced miRNAs are enriched in transiently expressed FLAG-AGOs compared to endogenous AGOs, this result suggests that cells might use uridylation to distinguish miRNAs at different stages of their metabolism.

### Coordination of TENT2, TUT4, and TUT7 in miRNA 3′ tailing

TENT2, TUT4, and TUT7 are the main TENTs implicated in miRNA tailing[41]. To comprehensively characterize their role in shaping miRNA sequence and abundance, we generated individual knockout, TUT4/TUT7 double knockout (DKO) and TENT2/TUT4/TUT7 triple knockout (TKO) cell lines by CRISPR. The expression of TENTs in WT cells and the loss of TENTs in corresponding KO cells were confirmed by Western blot (Fig. 2a). We performed AGO2-IP and deep sequenced AGO-associated miRNAs in all KO cells. Uridylation and adenylation on the 200 most highly expressed miRNAs were measured by calculating the frequency of the corresponding nucleotide in the NT tails.

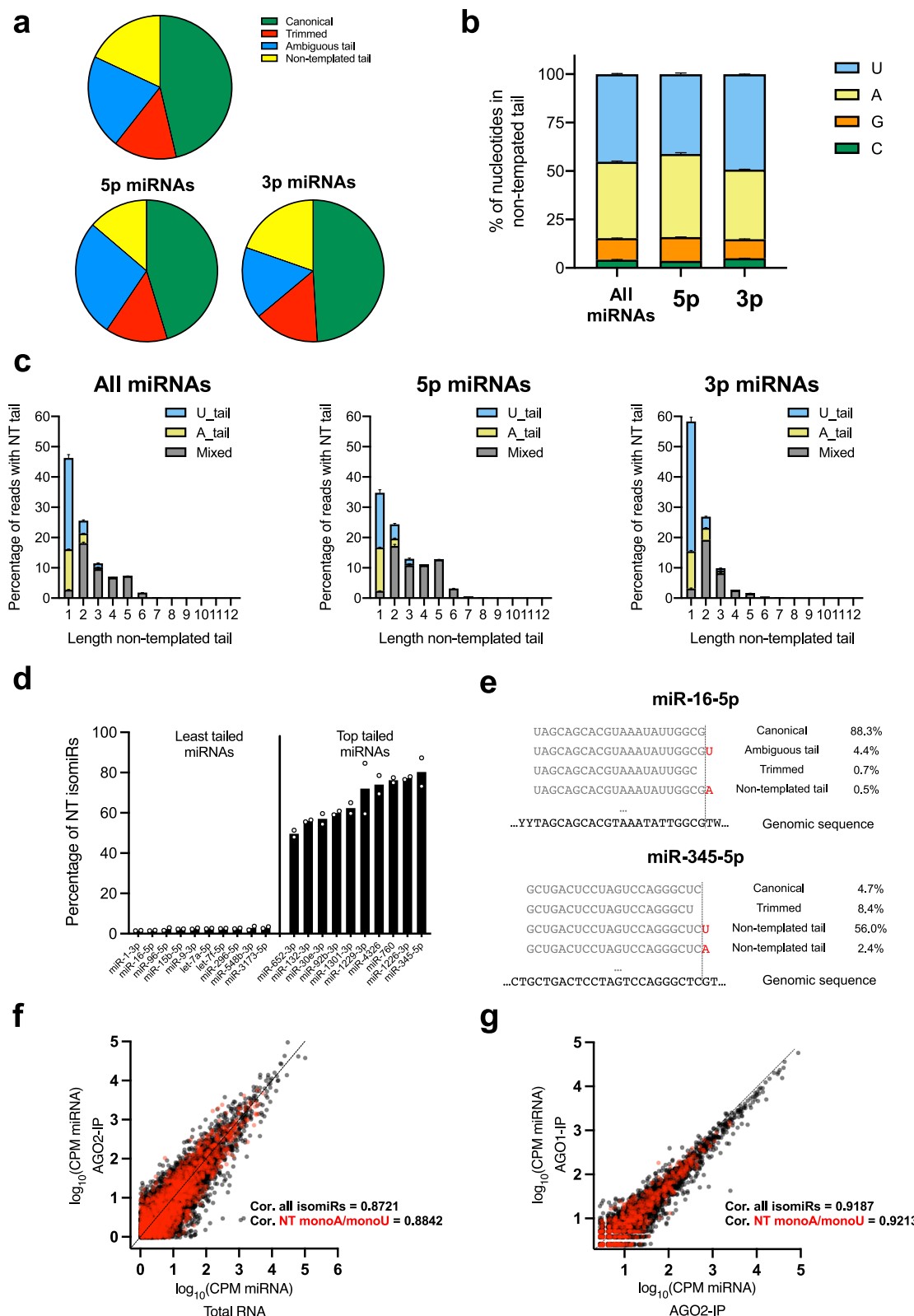

As expected, loss of TENT2 led to reduced adenylation, while loss of TUT4 and TUT7 resulted in reduced uridylation for both 5p and 3p miRNAs (Fig. 2b and Supplementary Fig. 2a). Decreased uridylation was accompanied by a moderate increase in adenylation (Fig. 2b), most likely due to improved accessibility of the miRNA 3′-end when the competing TENT was absent. As a result, abolishing either TUT4/7-mediated uridylation or TENT2-mediated adenylation did not

significantly diminish 3′ tailing. Substantial reduction of 3′ tailing and the resulting shortening of miRNA length was only observed in the TKO cells (Fig. 2b, c and Supplementary Fig. 2a, b). These results demonstrate that TENT2, TUT4, and TUT7 are the major TENTs that function coordinately in tailing mature miRNAs. Nonetheless, residual tailing observed in the TKO cells indicates that other TENTs, albeit to a lesser extent, also modify miRNA 3′-ends.

**Fig. 1 | miRNA 3′ tailing is frequent and specific.** Analyses of the top 200 most expressed miRNAs (top 100 5p and top 100 3p miRNAs). **a** Pie chart on the 3′-end composition (percentages). **b** Nucleotide identity on NT tails (average percentage ± standard error, $N = 200$ miRNAs). **c** Nucleotide identity on NT tails and tail length. Column height represents the percentage of NT tail while colored areas represent the corresponding fraction based on nucleotide identity. U_tail and A_tail represent uridylation and adenylation homogenous tailing, whereas Mixed includes heterogeneous tailing and other nucleotides (Average ± standard error, $N = 200$ miRNAs,

$N = 100$ 5p miRNAs and $N = 100$ 3p miRNAs). **d** Percentage of NT isomiRs for the 10 least and most tailed miRNAs ($N = 2$ biologically independent samples). **e** Examples of isomiR composition for miRNAs with the least (miR-16-5p) and the most (miR-345-5p) NT tailing. **f** Scatter plot of isomiR abundances between total RNA and AGO2-IP. **g** Scatter plot of isomiR abundances between AGO2-IP and AGO1-IP. Red dots indicate NT isomiRs with A or U tailing. Correlations were calculated using the Pearson correlation coefficient.

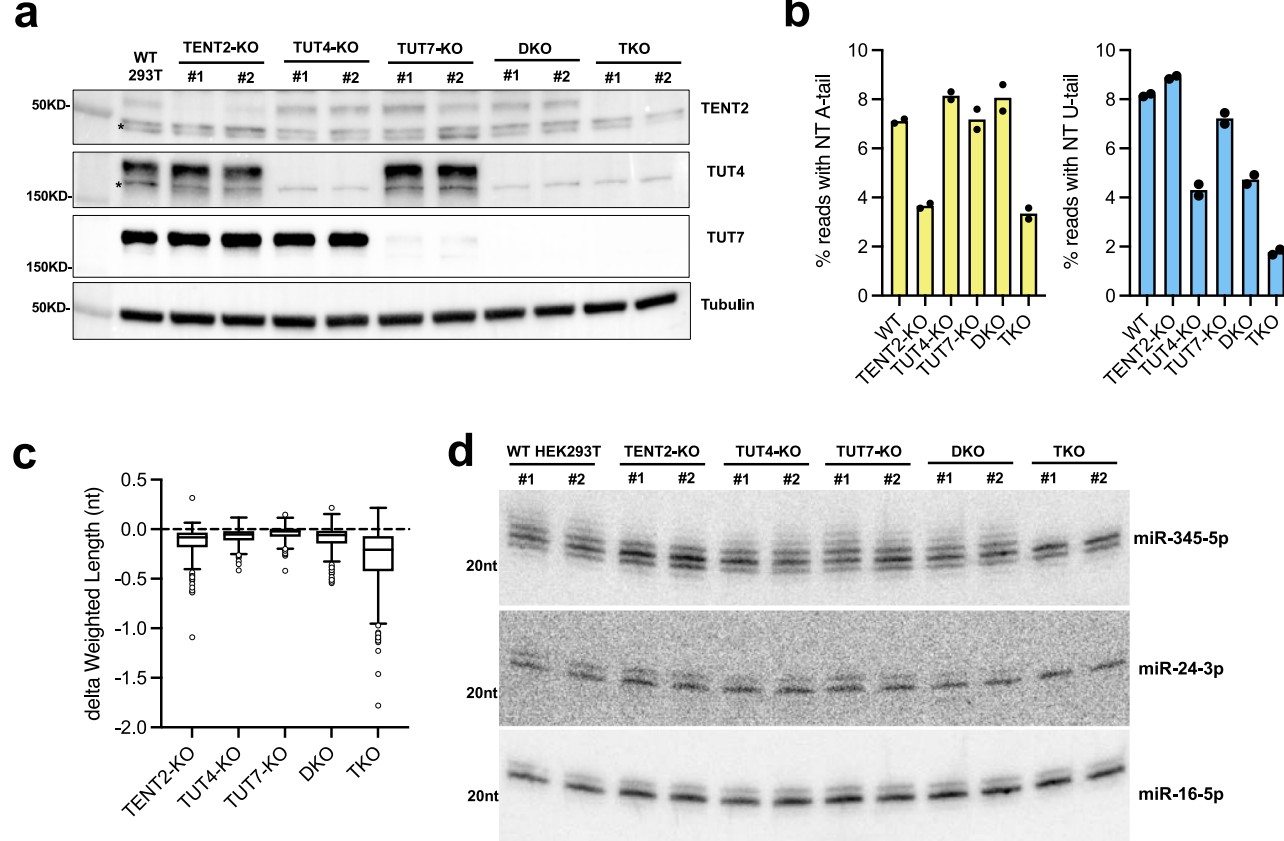

**Fig. 2 | Coordination of TENT2, TUT4, and TUT7 in miRNA 3′ tailing. a** Two clones were used for each isogenic cell line. TENT2, TUT4, and TUT7 were detected by western blot. Asterisks indicate non-specific bands. **b** The average percentages of NT A-tail (left) and U-tail (right) were shown in the figures. Percentage of NT tail was calculated by averaging top 200 expressed miRNAs ($N = 2$ biologically

independent samples). **c** Box-plot of the average length change (delta) between the different knockouts and WT cells (center median, whiskers based on Tukey test). Average lengths were weighted based on relative isomiR abundances within each miRNA ($N = 200$ miRNAs). **d** miR-345-5p, miR-24-3p, and miR-16-5p in each isogenic cell line were detected by Northern blot.

Further analyses on several individual miRNAs confirmed the conclusions drawn from the global analysis (Supplementary Fig. 2c). We validated these results by Northern blot (Fig. 2d). The tailed isomiRs that are longer than the cognate miRNAs run slower during electrophoresis. Consistent with the sequencing result (Fig. 1e), miR-345-5p was extensively tailed. These tailed isomiRs were diminished only in the TKO cells (Fig. 2d and Supplementary Fig. 2c–e), suggesting that TENT2, TUT4, and TUT7 are responsible for tailing miR-345-5p. By contrast, knockout of TENT2, TUT4, and TUT7 had no impact on the tailing of miR-16-5p (Fig. 2d), the tail of which are primarily templated (Fig. 1e and Supplementary Fig. 2c, e) and most likely are a result of alternative Dicer cleavage during biogenesis. Tailed isomiRs of miR-24-3p, which are mainly uridylated, were affected more by the knockout of TUT4/7 than that of TENT2 (Supplementary Fig. 2c–e). Tailing on miR-24-3p was greatly reduced in the DKO cells (Fig. 2d), indicating that TENT2 cannot fully complement the tailing mediated by TUT4/7. This suggests that adenylation and uridylation at miRNA 3′-ends are specific.

## TENT2 contributes to adenylation, uridylation, and guanylation of miRNAs

Our comprehensive set of knockout cells provided a unique opportunity to investigate TENT2-mediated miRNA tailing in vivo, which can be measured not only by comparing miRNAs between WT and TENT2-KO cells, but also by comparing the DKO and the TKO cells (Fig. 3a). With independent colonies serving as biological replicates, we were able to document consistent changes in miRNA tailing with high confidence. Furthermore, we performed the corresponding rescue assays with either a GFP control, WT TENT2, or catalytic dead (CD) mutant in both TENT2-KO and TKO cells (Supplementary Fig. 3a). Endogenous miRNAs were deep sequenced accordingly. Together with the KO datasets, these results allowed us to identify TENT2-specific tailing events with or without TUT4/7 in the background (Fig. 3a).

First, we sought to investigate the nucleotide specificity of TENT2. To this end, we examined TENT2-mediated adenylation by analyzing highly expressed isomiRs with an NT mono-A tail. As expected, the

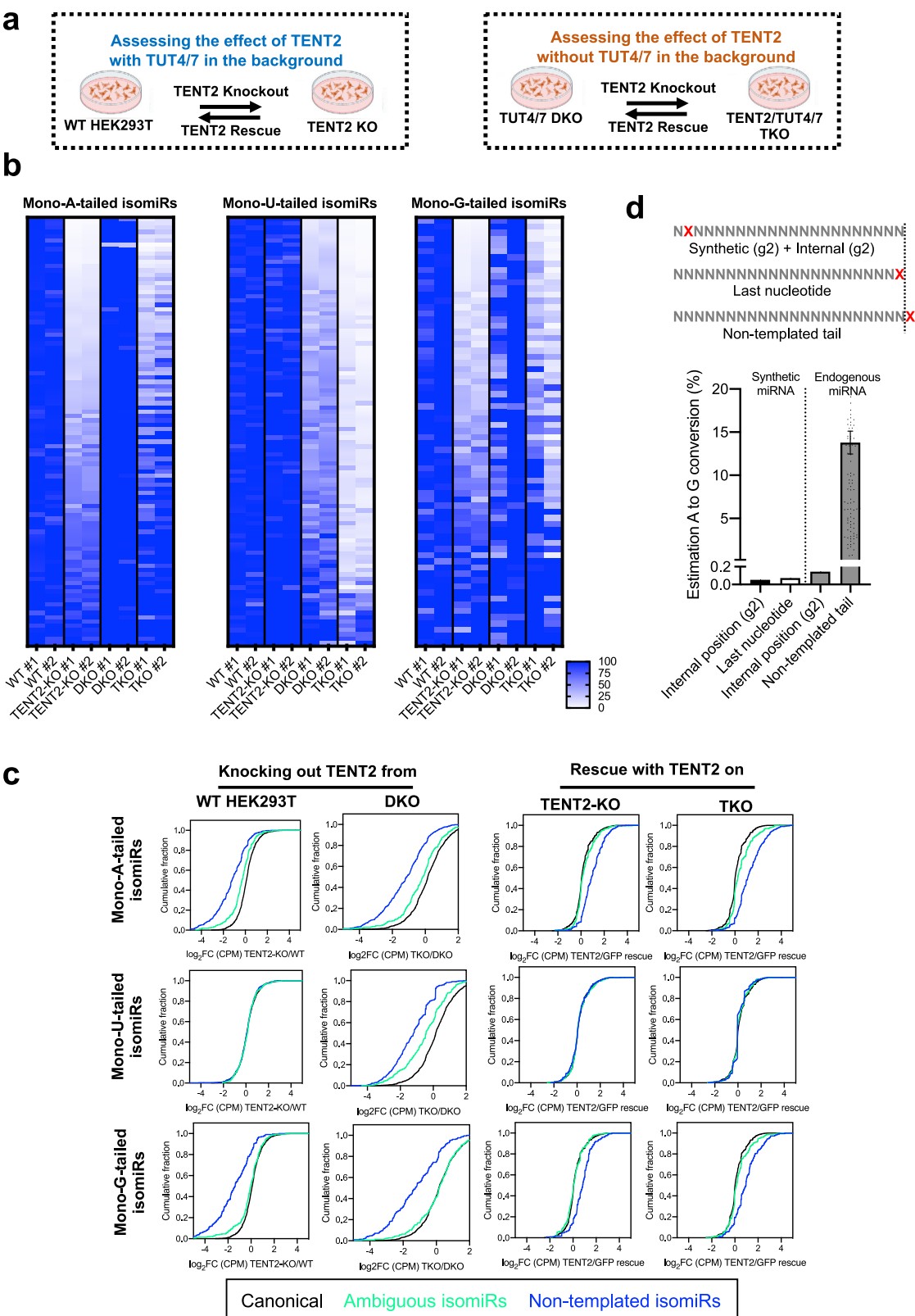

Fig. 3 | TENT2 contributes to adenylation, uridylation, and guanylation of miRNAs. a Scheme of the experimental strategy. Created with BioRender.com. b Heatmaps of the relative abundance of mono-tailed isomiRs (more details in Supplementary Fig. 3b) between WT and knockout cells. Colorimetric scale ranges in values between 0 and ≥100. c Cumulative curves of all mono-tailed isomiRs upon knockout or rescue of TENT2 in different cellular knockout backgrounds. Colored lines indicate canonical miRNAs (black), ambiguous (green), and NT (blue) isomiRs. d Scheme of the nucleotide positions analyzed. Estimated A to G conversions was calculated based on the known "A" identity of the sequence on synthetic (N = 11) or endogenous miRNAs (N = 100)(columns 1–3) and benchmarked against the observed abundance of NT A and G tails (column 4).

relative percentage as well as the absolute abundance of the mono-adenylated isomiRs decreased upon TENT2 knockout while knocking out TUT4 and TUT7 had no impact (Fig. 3b, c and Supplementary Fig. 3b). Expressing WT TENT2 but not the CD mutant in both TENT2-KO and TKO cells rescued the level of mono-adenylated isomiRs (Fig. 3c and Supplementary Fig. 3c, d). The abundance of isomiRs with an ambiguous mono-A tail also changed accordingly, albeit to a lesser extent (Fig. 3c). This is consistent with the notion that only a portion of isomiRs bearing ambiguous tails are a result of post-maturation modifications. IsomiRs adenylated with dinucleotide tails (AA, AU, and UA), although less abundant overall, followed a similar pattern of changes as mono-adenylated isomiRs (Supplementary Fig. 3e). Together, these results confirm the previously established adenylation activity of TENT2 on miRNA tailing, demonstrating the sensitivity and robustness of our approach.

We also examined the levels of uridylated isomiRs, which are not expected to be affected by TENT2. Consistent with previous studies, knocking out TENT2 in WT HEK293T cells had no impact on the level of mono-uridylated isomiRs (Fig. 3b, c). However, to our surprise, knocking out TENT2 on top of TUT4/7 knockouts led to a consistent and significant reduction ($p = 5.5e\text{-}127$) of uridylated isomiRs in both AGO2-associated and total miRNAs (Fig. 3b, c and Supplementary Fig. 3f, comparing DKO with TKO). This suggests TENT2-mediated uridylation activity that is masked by TUT4 and TUT7 in WT cells and only becomes observable when TUT4 and TUT7 are absent. Given that TENT2 shows negligible uridylation activity in vitro[42], this effect observed in cells is unlikely to be mediated by TENT2 directly. Rather, TENT2 may recruit additional TENT(s) other than TUT4/7 to uridylate miRNAs. Supporting this idea, while isomiRs with a dinucleotide A tail (AA) or U tail (UU) were dependent on TENT2 and TUT4/7 respectively, isomiRs with mixed dinucleotide tails (AU or UA) were more sensitive to the loss of TENT2 but less dependent on TUT4/7 (Supplementary Fig. 3e). This uridylation effect of TENT2, while evident during TENT2 knockout, was not rescued by expressing either the WT or the CD mutant of TENT2 in TKO cells (Fig. 3c, Supplementary Fig. 3c). It is possible that transient expression of TENT2 failed to reconstitute the potential complex between TENT2 and other uridylation TENT(s) in cells.

Interestingly, the levels of NT mono-G-tailed isomiRs also decreased upon TENT2 knockout and could be rescued specifically by the WT TENT2, mirroring the pattern observed for mono-A-tailed isomiRs (Fig. 3b, c, Supplementary Fig. 3c). This is specific to guanylated isomiRs since isomiRs with a NT mono-C-tail were hardly detectable and were not responsive to TENT2 knockout and rescue (Supplementary Fig. 3g). To rule out the possibility that the observed G-tail was a misread of A due to artifacts introduced during library construction and/or Illumina sequencing, we deep sequenced a set of synthetic miRNAs and calculated the presumable A to G conversion rate at both an internal position and at the last nucleotide. In both scenarios, the misread rate was below 0.1%, which is in line with the known misread rate of Illumina platforms (Fig. 3d). Furthermore, this presumably A to G conversion was not observed for endogenous miRNAs at an internal position. Rather, it was specific to the NT tail, supporting TENT2-mediated guanylation in miRNA tailing.

## TENT2 selectively modifies mature miRNAs but has minimal impact on miRNA abundance

While TENT2 knockout had a global impact on miRNA adenylation, different miRNAs responded differently (Fig. 3b). For miRNAs such as miR-615-3p and miR-149-5p, the NT adenylated isomiRs were almost completely gone in TENT2-KO and TKO cells (Fig. 4a and Supplementary Fig. 4a), indicating that TENT2 is solely responsible for the adenylation of this set of miRNAs. On the other hand, knocking out TENT2 in WT cells or DKO cells had marginal, if any impact on the adenylation of miRNAs such as miR-10a-5p (Fig. 4a), suggesting that

these miRNAs were adenylated by TENT(s) other than TENT2. Supporting this idea, the level of the mono-A tailed isomiR of miR-21-5p, which is generated by TENT4B[43], did not change after TENT2 loss (Supplementary Fig. 4a). NT adenylated isomiRs of other miRNAs, for example, let-7a-5p and miR-28-3p, were partially affected when TENT2 was knocked out (Fig. 4a and Supplementary Fig. 4a), indicating that both TENT2 and other TENT(s) contribute to their adenylation in HEK293T cells. In all scenarios, adenylated isomiRs increased in DKO cells, suggesting that TENT2 competes with TUT4/7 in accessing miRNA 3′-ends. Supporting this idea, TENT2 rescue effect was greater in TKO cells where TUT4/7 were absent than that in TENT2-KO cells (Fig. 4a and Supplementary Fig. 4a).

To further analyze the miRNA specificity of TENT2, we calculated the fold change of each NT mono-A-tailed isomiR upon TENT2 manipulation. We used it as the measurement for how responsive the corresponding miRNA is to TENT2-mediated modifications. The "TENT2 sensitivity" for each miRNA was not random but consistent when TENT2 was knocked out from two distinct genetic backgrounds (TENT2-KO vs. WT and TKO vs. DKO) (Fig. 4b). Furthermore, it was consistent between TENT2 loss and rescue (Fig. 4c). As expected, a stronger correlation between TENT2 knockout and rescue was observed when the competing TUT4/7 were absent. Using a similar approach, we examined the miRNA-specificity of TENT2-mediated guanylation and uridylation. Supporting the idea that both G-tail and A-tail are mediated by TENT2 directly, miRNAs showed correlated sensitivity between adenylation and guanylation upon TENT2 loss (Supplementary Fig. 4b). By contrast, there was no correlation between uridylation and adenylation, suggesting that the observed uridylation was mediated by TENT2 indirectly (Supplementary Fig. 4c). Finally, we measured miRNA specificity of TUT4/7-mediated uridylation (comparing DKO to WT cells) and found that it does not correlate with TENT2-mediated adenylation or uridylation (Fig. 4d), suggesting that the miRNA-specificity of TENT2 is unique and distinct from that of TUT4/7. Together, these results demonstrate that TENT2 is highly selective in tailing miRNAs.

Next, we sought to examine the impact of TENT2 on miRNA abundance by comparing miRNA levels in different KO cells. While knocking out TENT2 in HEK293T cells reduced the adenylation of most miRNAs, only miR-873-5p, miR-222-3p, miR-138-5p, and miR-34c-5p showed concurrent reduction in abundance (Fig. 4e). While this result supports the idea that these miRNAs are stabilized by TENT2-mediate adenylation, a mechanism established previously, further analyses indicate otherwise. Except miR-873-5p, these miRNAs showed a much smaller reduction when TENT2 was knocked out in the DKO cells than when TENT2 was knocked out in WT cells (Fig. 4f), suggesting a potential role that TUT4/7 played in the observed reduction. It is possible that TENT2 loss resulted in increased tailing by TUT4/7, which in turn led to miRNA decay. For miR-873-5p, the level of its passenger strand (miR-873-3p) was also reduced when TENT2 was abolished (Supplementary Fig. 4d). This suggests that TENT2 regulates miR-873 at a stage before the separation of guide and passenger strands, likely at the pri-miRNA level. Supporting this idea, TENT2 knockout also decreased the level of miR-876-5p (Supplementary Fig. 4e), which shares the same pri-miRNA transcript with miR-873. Furthermore, expressing TENT2 in either TENT2-KO or TKO cells failed to rescue the reduction of these miRNAs including miR-873 (Fig. 4g). Together, these results suggest that TENT2 has only marginal impacts on miRNA abundance in HEK293T cells.

## TUT4 but not TUT7 selectively uridylates most mature miRNAs

Applying a similar strategy, we investigated TUT4/7-mediated miRNA tailing by comparing DKO cells to WT cells as well as by comparing TKO cells to TENT2-KO cells (Supplementary Fig. 5a). Together with TUT4/7 rescue (Supplementary Fig. 5b), we were able to identify TUT4/7-specific effects. As expected, losing TUT4 and TUT7 led to decreased

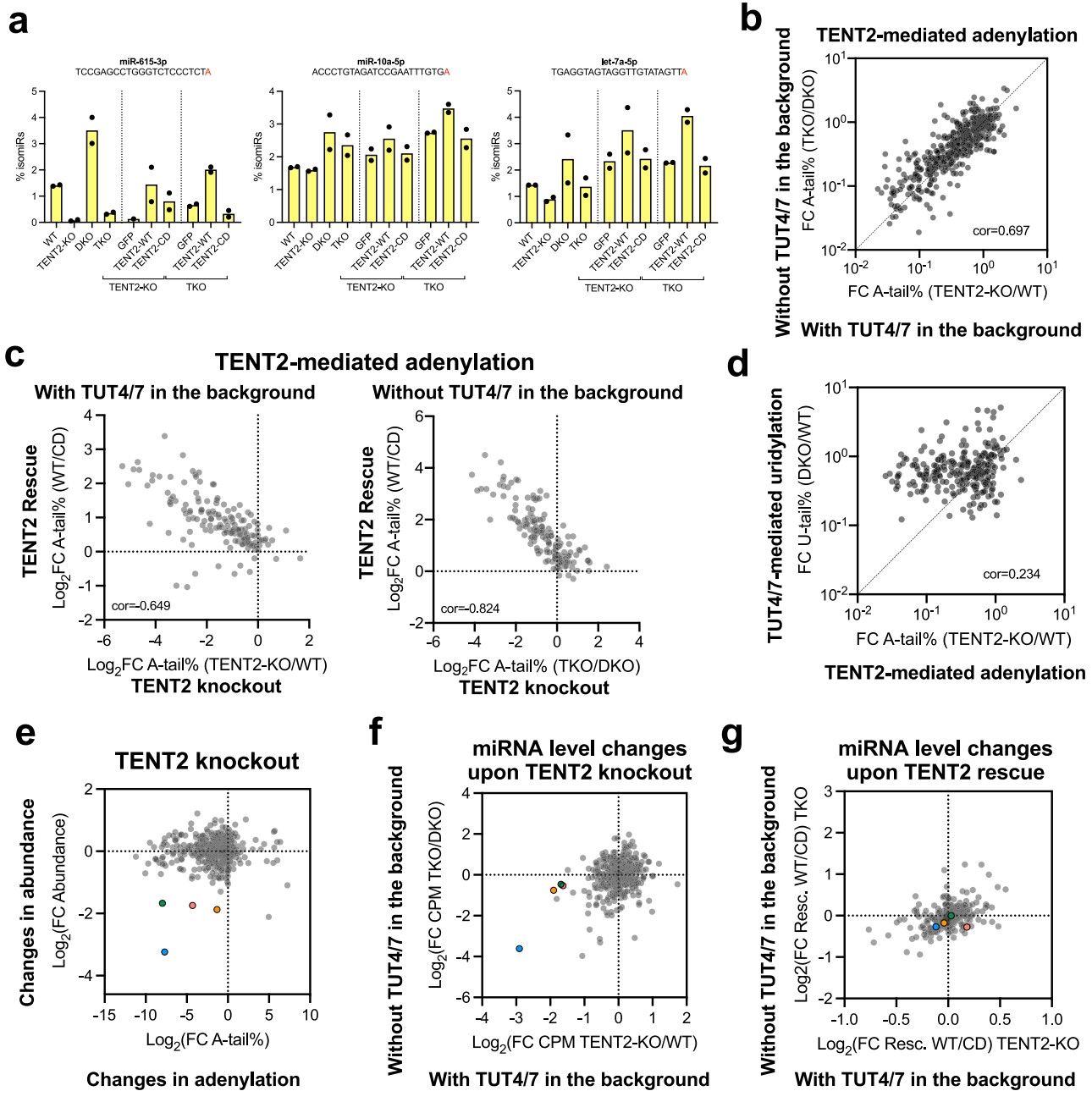

**Fig. 4 | TENT2 selectively modifies mature miRNAs but has minimal impact on miRNA abundance. a** Percentage of adenylated isomiRs across different knockouts and rescues. Sequences indicate templated nucleotides (black) and NT adenylation (red). Dots represent the values obtained in the two independent colonies or biological replicates. **b, c** Scatter plot of fold-change in the percentage of NT A-tail between different genetic backgrounds and rescues. **d** Scatter plot of fold-change in the percentage of NT A-tail and U-tail (matched isomiRs) between different genetic backgrounds. R was calculated using the Pearson correlation coefficient. **e** Scatter plot of fold-changes in the percentage of NT A-tail (*x* axis) and overall miRNA abundance (*y* axis) upon TENT2-KO. **f, g** Scatter plot of fold-changes in overall miRNA abundance upon TENT2 knockout or rescue in different genetic backgrounds. Dots highlighted correspond to miR-34c-5p (green), miR-138-5p (red), miR-222-3p (orange), and miR-873-5p (blue).

levels of mono-U-tailed isomiRs (Supplementary Fig. 5c, d). Knocking out TUT4 and TUT7 caused a subtle increase in the levels of mono-A-tailed isomiRs, demonstrating again the competition in accessing miRNA 3′-ends between uridylating and adenylating TENTs (Supplementary Fig. 5d). Consistent with the idea that TENT2 also contributes to miRNA uridylation, knocking out TUT4/7 in the absence of TENT2 (TKO vs. TENT2-KO) had a much larger effect in reducing miRNA uridylation than the same procedure with TENT2 in the background (DKO vs. WT) (Supplementary Fig. 5e). Ectopically expressing both TUT4 and

TUT7 from plasmids in the KO cells rescued miRNA uridylation but had no impact on adenylation (Supplementary Fig. 5d, f), indicating that, different from TENT2, TUT4/7 only exert uridylation activity in cells.

So far, we have evaluated the combined activity of TUT4 and TUT7. Taking advantage of TUT4-KO and TUT7-KO cells, we examined their activity individually. We analyzed the 100 most abundant uridylated isomiRs detected in these cells and calculated their relative abundance compared to that in the WT cells. To our surprise, while both TUT4 and TUT7 were expressed in HEK293T cells

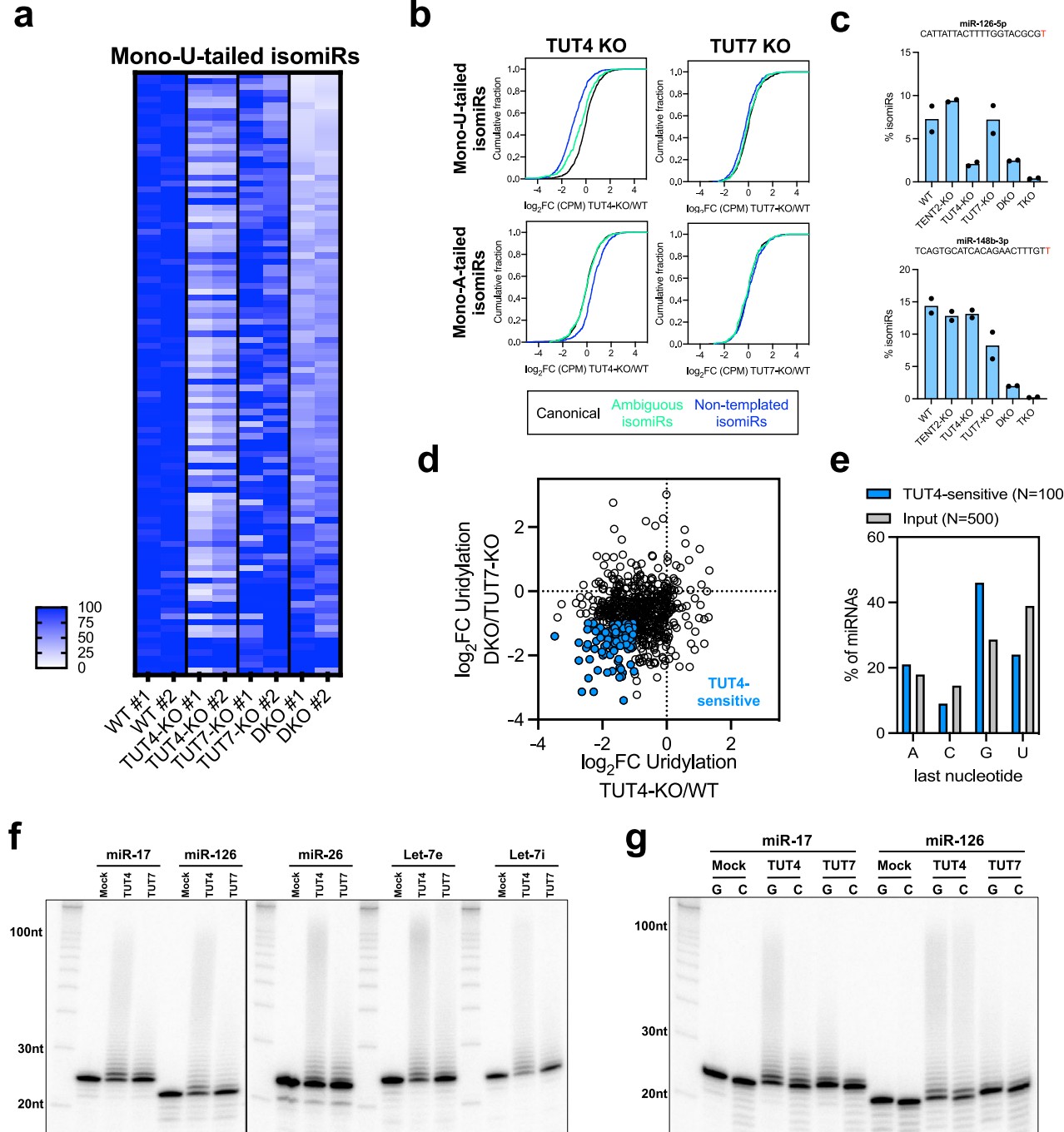

**Fig. 5 | TUT4 but not TUT7 selectively uridylates most mature miRNAs.**
**a** Heatmap of the relative abundance of mono-uridylated isomiRs between WT and knockout cells. Colorimetric scale ranges in values between 0 and ≥100 (more details in Supplementary Fig. 5c). **b** Cumulative curves of all mono-uridylated (upper panel) and mono-adenylated (lower panel) isomiRs upon knockout of TUT4 or TUT7. Colored lines indicate canonical miRNAs (black), ambiguous (green), and NT (blue) isomiRs. **c** Percentage of uridylated isomiRs across single, double, and triple knockout of TENT2, TUT4, and TUT7. Sequences indicate templated nucleotides (black) and NT adenylation (red). Dots represent the values obtained in the two independent colonies. **d** Scatter plot of fold-changes in the percentage of NT U-tail upon the knockout of TUT4 in different genetic backgrounds. Dots highlighted in blue indicate a subset of uridylated isomiRs more sensitive to TUT4. **e** Analysis of TUT4 tailing preference based on the last templated nucleotide preceding the uridylation. The TUT4-sensitive group ($N = 100$) was defined based on the TUT4 knockout response described in Fig. 5d. **f** In vitro uridylation of single-stranded miRNA by purified TUT4 or TUT7 protein. **g** In vitro uridylation of single-stranded miRNA where the last nucleotide has been changed from "G" to "C".

(Supplementary Fig. 5b), losing TUT4 had a much bigger impact on miRNA uridylation than losing TUT7 (Fig. 5a and Supplementary Fig. 5c, g). The absolute levels of uridylated isomiRs followed the same pattern (Fig. 5b and Supplementary Fig. 5h). For most miRNAs, such as miR-126-5p, the level of mono-U-tailed isomiRs in TUT4-KO cells resembled that in DKO cells. Only a small portion of miRNAs (15 out of

top 100 expressed, Supplementary Table 1) such as miR-148b-3p were affected by TUT7 knockout more than that by TUT4 knockout (Fig. 5c). Interestingly, 14 of these 15 miRNAs are 3p miRNAs, the uridylation of which could be inherited from pre-miRNAs (Supplementary Fig. 1b). These results suggest that TUT4 is responsible for uridylation of mature miRNAs whereas TUT7-mediated uridylation is limited to a

subset of miRNAs and potentially occurs only at the pre-miRNA stage. This is unlikely due to the inability of TUT7 to uridylate mature miR-NAs, because ectopically expressing TUT7 in DKO or TKO cells had a similar, if not stronger effect in rescuing miRNA uridylation than expressing TUT4 (Supplementary Fig. 5i).

Next, we sought to examine the miRNA-specificity of TUT4 by identifying miRNAs whose uridylation was relatively more dependent on TUT4. To this end, we measured the fold change of the uridylation percentage for each miRNA upon TUT4 knockout in two comparisons (TUT4-KO vs. WT and DKO vs. TUT7-KO). The top 100 miRNAs with the biggest changes in uridylation in both scenarios were cataloged as "TUT4-sensitive" (Fig. 5d). Interestingly, G was enriched whereas U and C were disfavored at the last nucleotide position in these "TUT4-sensitive" miRNAs (Fig. 5e). We also probed the specificity of TUT4 by analyzing the rescue dataset. While ectopically expressing either TUT4 or TU7 rescued the loss of uridylation in both DKO and TKO cells, different miRNAs responded differently (Supplementary Fig. 5j), indicating they have distinct specificity. Once again, G was enriched while U and C were disfavored as the ending nucleotide in the top 100 miRNAs that were consistently more responsive to TUT4 rescue than TUT7 rescue in both DKO and TKO cells (Supplementary Fig. 5k). Together, these results suggest that the ending nucleotide plays a role in determining the miRNA specificity of TUT4.

To investigate the underlying mechanisms of these observations, we performed an in vitro uridylation assay. Mature let-7i (22 nt) was chemically synthesized, labeled with P32 at the 5′-end, and incubated with either TUT4 or TUT7 protein. The products were separated on a PAGE gel. As expected, both TUT4 and TUT7 uridylated mature let-7i, resulting in isomiRs with U-tail, the length of which depended on the amount of enzyme and UTP/substrate ratio (Supplementary Fig. 5l). We performed in vitro uridylation assays on five miRNAs with the same amount of TUT4 and TUT7 proteins (Supplementary Fig. 5m). Regardless of the miRNA sequence, TUT4 showed more robust activity than TUT7 (Fig. 5f and Supplementary Fig. 5n), partially explaining the superior activity of TUT4 observed in cells. However, miRNAs ending with a G (miR-17-5p and miR-126-5p) were uridylated by TUT4 with similar efficiency as miRNAs ending with a U (miR-26, let-7e, and let-7i) (Fig. 5f and Supplementary Fig. 5n), despite the fact that only the former group was efficiently uridylated in cells. Furthermore, while mutating the last G of miR-17-5p to a C reduced its uridylation efficiency by TUT4 and TUT7, the same G to C mutation of miR-126-5p did not significantly affect its uridylation by either TUT4 or TUT7 (Fig. 5g and Supplementary Fig. 5o). Together, these results show that TUT4's miRNA preference cannot be fully explained by the intrinsic feature of the enzyme, but is rather specific to miRNA tailing in cells.

## TUT4/7 regulate the abundance of a set of miRNAs via distinct mechanisms

Finally, we examined the impact of uridylation on miRNA abundance by comparing miRNA levels in KO cells to those in WT cells. Consistent with the model that TUT4 and TUT7 negatively regulate the biogenesis of let-7, we found let-7 family members among the most upregulated miRNAs in DKO and TKO cells (Fig. 6a). A previous study showed that a subclass of let-7 family members (let-7b, let-7d, let-7g, let-7i, miR-98) are more responsive to regulation via the LIN28-TUT-DIS3L2 machinery due to stronger binding between LIN28 and their precursors[44]. Only this subset but not the other let-7 family members were identified as the most upregulated miRNAs in DKO and TKO cells, demonstrating the robustness and specificity of our analysis. Several mirtrons including miR-1229-3p and miR-3620-3p were also upregulated, suggesting that TENT-mediated uridylation may regulate mirtron biogenesis in mammals as well. Consistent with our previous finding that mature miR-222-3p is targeted by the TUT-DIS3L2 machinery[33], miR-222-3p abundance increased upon TUT4/7 knockout (Fig. 6a).

TUT4 and TUT7 individual KO cells presented a relatively small number of differentially expressed miRNAs, while DKO cells had more and the TKO cells had the most (Supplementary Fig. 6a), a pattern similar to the observed effect of TENT-mediated uridylation on mature miRNAs (Fig. 5a and Supplementary Fig. 5h). While this result supports that uridylation regulates miRNA abundance, the fact that both upregulated and downregulated miRNAs were observed in each KO cell line indicates that the underlying mechanisms are complicated. Indeed, further analysis on individual miRNAs showed no correlation between the changes in 3′ mono-uridylation and the alteration in abundance upon TENT loss (Supplementary Fig. 6b, c), indicating that 3′ uridylation is unlikely to be a ubiquitous pathway for miRNA decay. Nonetheless, we were able to systematically identify miRNAs regulated by TUT4 and/or TUT7 by comparing miRNA abundance upon TUT4/7 knockout and the corresponding rescue (Fig. 6b). We focused on miRNA level changes that were consistently observed in DKO and TKO cells and could be rescued by ectopic expression of TUT4 and/or TUT7. Using this method, we identified a set of miRNAs that were either positively or negatively regulated by TUT4 and TUT7 in HEK293T cells with high confidence (Fig. 6c and Supplementary Table 1).

Besides let-7 family members, our analysis revealed miR-181b-5p as one of the miRNAs negatively regulated by TUT4/7. We focused on miR-181b-5p due to its known function in cell proliferation and tumorigenesis[45]. We validated the sequencing results by Northern blot. Similar to let-7i-5p, let-7g-5p, and miR-222-3p, miR-181b-5p level was upregulated upon TUT4/7 knockout while the control miR-16-5p level remained unchanged (Fig. 6d). Furthermore, expressing TUT4, TUT7 or both, but not the GFP control in both DKO and TKO cells led to a specific downregulation (Fig. 6e). Interestingly, miR-181a-5p, a miRNA clustered with miR-181b-5p on the same pri-miRNA transcripts, followed a similar pattern of changes albeit to a smaller extent, suggesting that part of the regulation occurs at the pri-miRNA level. By contrast, miR-221-3p which is generated from the same pri-miRNA transcript as miR-222-3p, showed a distinctive pattern of changes, indicating that TUT4/7-mediated downregulation of miR-222-3p is post-transcriptional. Finally, to check if TUT4/7 regulates miR-222-3p and miR-181b-5p levels in the same way as let-7, we knocked down LIN28B, the LIN28 isoform expressed in HEK293T cells by siRNAs in WT, DKO and TKO cells (Fig. 6f). As expected, let-7g-5p was upregulated upon LIN28B knockdown in a TUT4/7-dependent manner. By contrast, neither miR-181b-5p nor miR-222-3p was affected by LIN28B knockdown with or without the presence of TUT4/7, indicating that their regulation is independent of LIN28B.

Surprisingly, we found that TUT4 and/or TUT7 positively regulate a set of miRNAs. However, most of these miRNAs were either lowly expressed or had only subtle changes upon TUT4/7 knockout and/or rescue. Unlike TUT4/7-mediated downregulation, the upregulation effect observed in deep sequencing could not be fully validated (Supplementary Fig. 6d). Nonetheless, we focused on those supported by Northern blot and found that miR-888-5p and miR-892a-3p were dramatically upregulated when TUT7 was ectopically expressed in either DKO or TKO cells (Supplementary Fig. 6e). Wild type TUT7, but not the CD mutant (TUT7-CD), resulted in the upregulation (Fig. 6g), indicating that uridylation activity is required. miR-888-5p and miR-892a-3p are two of seven miRNAs that reside within a human-specific cluster on chromosome X (Supplementary Fig. 6f)[46]. This strongly suggests that TUT7 led to the upregulation of the pri-miRNA transcripts. Supporting this idea, all seven miRNAs from the miR-888 cluster showed the same pattern of changes upon TUT4/7 knockout and rescue (Supplementary Fig. 6g), indicating that TUT7-mediated uridylation either promotes the transcription or stabilizes the pri-miR-888 cluster transcript. Together, these results demonstrate that TUT4 and/or TUT7 regulate miRNAs including let-7, miR-222, miR-181b, and miR-888 cluster by distinctive mechanisms.

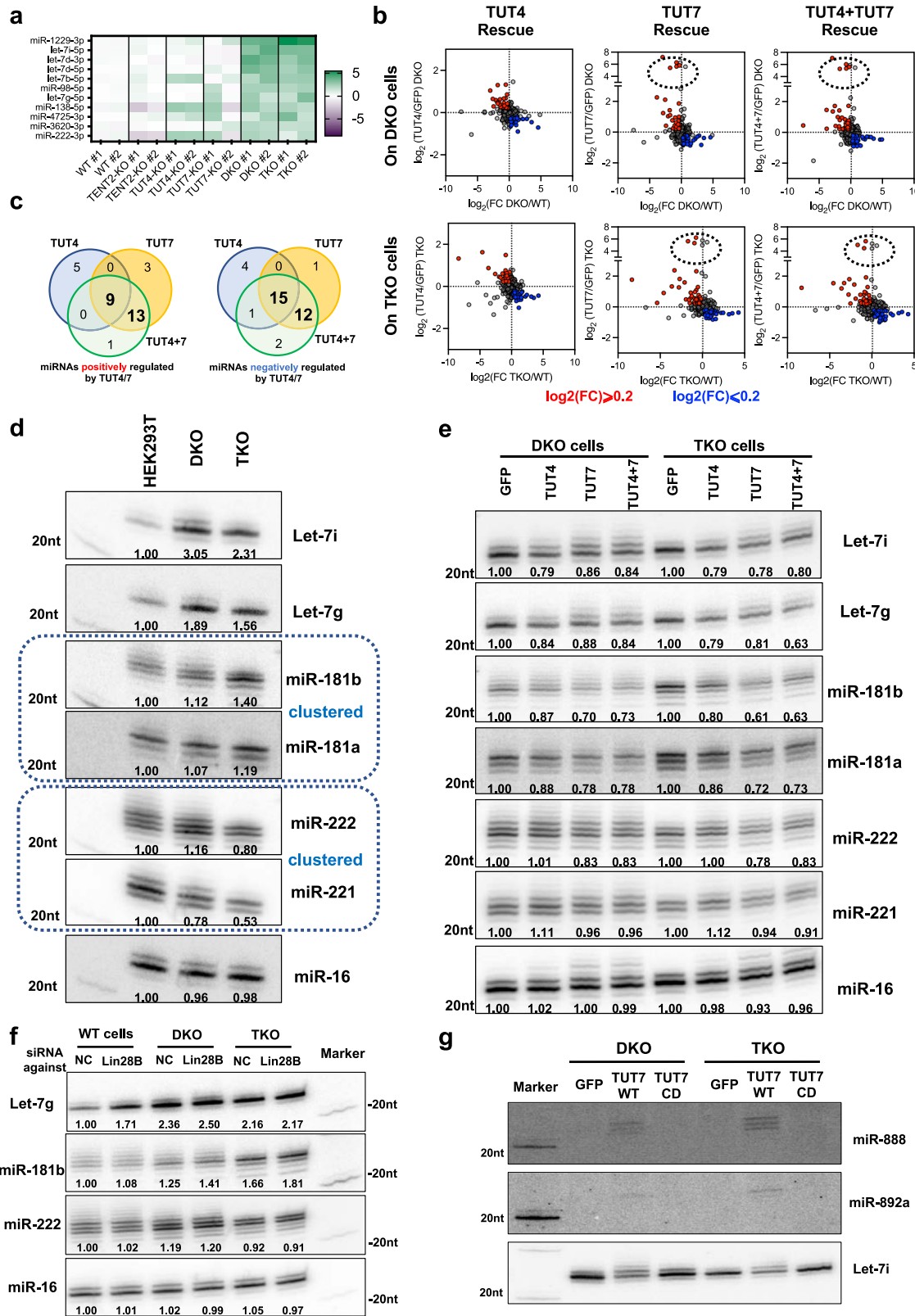

**Fig. 6 | TUT4/7 regulate the abundance of a set of miRNAs via distinct mechanisms. a** Heatmap illustrating the miRNAs with the highest expression increase upon TENT2, TUT4, and TUT7 knockouts. **b** Analysis comparing the effects on miRNA abundance upon knockout and rescue with TUT4, TUT7, or TUT4/7 in different genetic backgrounds. Colored dots indicate log2(FC)l ≥ 0.2 (upregulation, red) and log2(FC) ≤ 0.2 (downregulation, blue). Dashed circle indicates members of the miR-888 cluster. **c** Venn diagram illustrating miRNAs positively (left panel) and negatively (right panel) regulated by TUT4/7. This diagram was based on the

consensus responses observed in Fig. 6b. **d** Northern blot validating the level changes of TUT4 and TUT7 regulated miRNAs in WT, DKO, and TKO cells. **e** Northern blot validating TUT4 and TUT7 rescue effects on the levels of TUT4 and TUT7 regulated miRNAs. **f** Northern blot detecting miRNA levels in WT, DKO, and TKO cells transfected with control or LIN28B siRNA. **g** The levels of miR-888 and miR-892a in the DKO and TKO cells upon rescue with TUT7-WT or TUT7-CD were detected by Northern blot, let-7i served as a control.

## TUT4 and TUT7 promote growth of the HEK293T cells

Finally, we sought to investigate the biological function of TENT2, TUT4 and TUT7 by examining the impact of their knockouts on cell proliferation. To this end, we measured the time it takes for cell number to double before reaching confluency. For each KO cell line, 8 replicates consisting of two independent colonies were tested. While all knockouts grew at a slower rate compared to that of the WT cells, a significant difference ($p < 0.01$) was only observed when all three TENTs were lost (Supplementary Fig. 7a). To rule out the possibility that the reduced cell growth was due to off-target effects of genome editing, we performed rescue assay with either TENT2, TUT4, or TUT7 in TKO cells. Compared to the expression of the CD mutants, expression of TUT4 or TUT7 increased cell growth rate, whereas expression of TENT2 had opposite, if any, effect on cell proliferation (Supplementary Fig. 7b). Together, these results indicate that uridylation mediated by TUT4 and TUT7 promotes cell growth of HEK293T cells. This is consistent with previous studies performed in multiple cancer cell lines, suggesting a general role of TUT4/7 in cell proliferation[34,35,47,48]. Nonetheless, future studies are required to illustrate how exactly cell proliferation is affected and more importantly, the role that miRNAs play in this process.

## Discussion

It has become apparent that 3′ tailing is an integral mechanism in regulating miRNA function. A major challenge in studying miRNA tailing is the lack of understanding of the selective nature of TENT actions. Here, we used a genetic approach to tackle this question. Different from previous efforts, we generated a comprehensive set of single, double, and triple KO cell lines, which enabled us to determine the functions of TENT2, TUT4, and TUT7 by at least two distinct genetic comparisons. Together with the rescue assays, we identified TENT-specific effects, some of which are too subtle to have been documented previously. We sequenced 38 samples, generating more than 180 million miRNA reads. Analyses of these sequencing results were further validated by Northern blots and in vitro assays. This comprehensive approach allows us to make novel findings that advance our understanding of how miRNAs are regulated by TENTs.

The role of TENT2 as an adenylation enzyme was well-established. Surprisingly, we found that human TENT2 also possesses guanylation activity in cells, which is consistent with a recent study where human TENT2 activity was measured by deep sequencing when tethered to a tRNA in yeast[49]. The resulting G-tail on miRNAs is unlikely an artifact because: (1) It was not observed on synthetic miRNA spike-in. (2) Knocking out TENT2, but not TUT4 and/or TUT7, led to the decrease of G-tail. (3) The reduction of G-tail in TENT2-KO cells was rescued by ectopically expressing the WT but not the CD mutant of the TENT2. Nonetheless, the guanylation activity of TENT2 is barely detectable in vitro[42], suggesting that it could be enhanced by cellular cofactors. TENT4A and TENT4B, two adenylation enzymes, also have guanylation activity, which plays an important role in regulating mRNA turnover[50]. It is possible that this "mix-tailing" is a general feature of adenylating TENTs. The functional impact of TENT2 guanylation is yet to be determined. Another unexpected observation is that TENT2 also contributes to the uridylation of miRNAs. Since this effect is only observable when TUT4 and TUT7 are absent, it remains to be examined whether TENT2 mediates uridylation in WT cells or if it is only a compensation mechanism to maintain uridylation on miRNAs. Nonetheless, our results indicate that TENT2, in addition to TUT4/7, should be considered when investigating the function of uridylation on miRNAs.

Different from the nucleotide preference, little is known regarding the TENT preference towards various miRNA sequences in cells. Taking advantage of our comprehensive set of isogenic TENT KO cell lines as well as the corresponding rescue assays, we found that the 3′ modifications mediated by TENT2, TUT4, and TUT7 are highly specific. Although TENT2-mediated adenylation and TUT4/7-mediated uridylation compensate each other to a certain extent, their specificity towards different miRNAs is distinctive and consistent among different KO cells. Given that TENT2 lacks an RNA-binding domain, cellular cofactors such as QKI-7 likely account for its miRNA-specificity[51,52]. TUT4 and TUT7, on the other hand, can uridylate ssRNA in vitro without RNA-binding cofactors. *Drosophila* TUT enzyme Tailor prefers substrates ending with guanosine or uridine nucleotides in vitro[27,53]. Similarly, we found that human TUT4 and TUT7 uridylate G-ending and U-ending miRNAs with an efficiency higher than C-ending miRNAs in vitro, suggesting that this intrinsic feature of the TUT enzyme is conserved. However, consistent with a previous study[54], their in-cell miRNA-specificity cannot be fully explained by their substrate preference measured in vitro. Together, these results suggest that 3′ tailing on miRNAs is strictly regulated in cells. Given that the 3′-end of a mature miRNA is inserted inside the PAZ domain of AGO2[55], the accessibility of the 3′-end likely plays a role in regulating 3′ tailing. It is possible that 3′ modifications occur after miRNA pairing with its targets. In this way, the type of pairing and/or flanking sequences of the target site underlie the specificity of tailing on the corresponding miRNA.

One of the best-characterized functions of TENT2, TUT4, and TUT7 is regulating miRNA abundance, which is highlighted by their prominent roles in the biogenesis of let-7 family members[56]. By contrast, whether or not miRNA tailing determines miRNA stability remains debatable. Consistent with several recent studies[35,57,58], we found that the loss of uridylation or adenylation did not coincide with changes in miRNA abundance at a global scale. This suggests that 3′ tailing does not play a general role in miRNA turnover. Rather, TENT2 stabilizes certain miRNAs in a context-dependent manner, while TUT4 and TUT7 regulate the abundance of a subset of miRNAs via multiple mechanisms. We found that TUT4 and TUT7 negatively regulate miR-222-3p but not the clustered miR-221-3p independently of LIN28B, indicating that the regulation likely occurs at the mature miRNA level. This is consistent with our previous observation that TUT4/7-mediated oligo-uridylation was specifically observed on miR-222-3p when DIS3L2 was knocked out[33]. Together, these results support a model in which AGO2-associated mature miR-222-3p was targeted by the TUT-DIS3L2 machinery. Future studies are required to understand why miR-222-3p is specifically targeted in HEK293T cells. We also identified miR-181b-5p and miR-888 cluster miRNAs among others regulated by TUT4/7, indicating that the regulation is not limited to the biogenesis of let-7 family members. Nonetheless, the fact that only a small set of miRNAs are subjected to TENT regulation suggests that the biological functions of the 3′ modifications may lie beyond controlling miRNA abundance.

TUT4 and TUT7 are thought to act redundantly. Our findings challenge this model by showing that TUT4 and TUT7 possess different tailing activities towards miRNAs. Knocking out TUT4 had a major impact on miRNA uridylation overall, whereas the influence of knocking out TUT7 was limited to a subset of 3p miRNAs. These results suggest that, while both contribute to the uridylation of pre-miRNAs, TUT4 is the major enzyme tailing mature miRNAs. Knocking out TENT2 further reduced miRNA uridylation (Supplementary Fig. 5h), suggesting that there might be a hierarchical preference among uridylation mediated by TUT4, TUT7, and TENT2. We have previously found that TUT7 is more robust in oligo-uridylation of mature miRNAs, which are subsequently degraded by DIS3L2[33]. These results suggest a model in which TUT4 is mainly responsible for mono-uridylation while TUT7 is involved in oligo-uridylation, leading to different outcomes: the former generates uridylated isomiRs, which might possess altered target repertoire[19], while the latter results in degradation. In this way, cells distinguish these two pathways by functional specialization of TUT4 and TUT7. Supporting this idea, TUT7 but not TUT4 plays a major role in degrading the mRNAs of histone and ZC3H12a[59,60]. Furthermore, the

upregulation of miR-888 cluster miRNAs, most likely due to the downregulation of unknown factors, is specific to TUT7. Further studies are required to understand the underlying mechanism for this functional specialization and how the specificity is achieved.

# Methods

## Plasmids

For TENT2 gRNA expression vectors, 4 gRNAs were designed and synthesized, then cloned into Lenti-CRISPR-V2-puro at BsmbI sites. TENT2 gRNA3-gRNA2/pTER+ plasmid was constructed by sequentially inserting the PCR amplified "U6-gRNA-sgRNA scaffold-TTTTTT" fragment into pTER+ at BglII and HindIII sites. pIRESneo-FLAG/HA AGO2 is from Addgene (#10822). The coding sequence of TENE2, TUT4, and TUT7 were PCR amplified from a pool of HEK293T cDNAs and then cloned into pIRESneo-FLAG/HA at NheI and EcoRI sites using In-fusion HD kit (Clontech, 638911). The CD point mutations in pIRESneo-FLAG/HA-TENT2-CD (D212A and D214A), pIRESneo-FLAG/HA-TUT4-CD (D1009A and D1011A), and pIRESneo-FLAG/HA-TUT7-CD (D1058A and D1060A) were introduced by mutagenesis[31,61]. All the oligo and primer sequences are listed in Supplementary Table 2.

## Cell lines

HEK293T cells were maintained in DMEM high glucose (Gibco, 11995-073) supplemented with 10% heat-inactivated fetal bovine serum (Hyclone, SH30070.03HI), 100 U/ml penicillin-streptomycin (Gibco, 15140163) at 37 °C. TENT2-KO cells, TUT4-KO cells, and TUT7-KO cells were generated by the lenti-CRISPR V2 system using four corresponding gRNAs. TKO cells were generated by transfecting TENT2 gRNA3-gRNA2/pTER+ into DKO cells and selection with Zeocine. Plasmid transfections were performed using PolyJet™ DNA Transfection Reagent (SignaGen, SL100688-5) according to the manufacturer's instructions. Lipofectermine RNAimax (Thermo scientific, 13778150) was used for LIN28B siRNA (on-target smart pool, Dharmacon, L-028584-01-0005) transfections with a reverse transfection protocol.

## Cell growth rate measurement

To measure cell growth in real-time, the xCELLigence Real-Time Cell Analysis (RTCA) DP instrument (Agilent) was used to noninvasively monitor the viability of cultured cells using electrical impedance as the readout at 37 °C with 5% CO2. 5000 cells were seeded into each well of an E-plate (gold microelectrodes embedded at the bottom of 16 well microplates, Agilent, 5469830001) with four replicates for each cell line or treatment. The impedance was recorded at 15 min intervals.

## Northern blotting

Total RNA was isolated from cells using Trizol (Life Technologies,15596-018) and quantitated by Nanodrop. 20 μg total RNA was run on 20% (w/v) acrylamide/8 M urea gels with a $^{32}$P-labeled Decade marker (Ambion, AM7778), and then transferred onto Hybond-N membranes (Amersham Pharmacia Biotech, RPN303N). After transfer, the membrane was either UV crosslinked or EDC-mediated chemical cross-linking was used. $^{32}$P-labeled probes that reverse complement to the targeted miRNAs were hybridized with membrane in PerfectHyb™ Plus Hybridization Buffer (Sigma, H7033) overnight at 37 °C. After washing with 2× SSC with 0.1% SDS buffer for 3 × 15 min at 37 °C, the membrane was exposed to an Imaging Screen-K overnight. Images were then analyzed by the Typhoon Trio Imaging System. NB results were processed and quantitated by Image J software 1.52a.

## IP for endogenous and ectopic AGO1 and AGO2-associated miRNAs

For endogenous AGO1 and AGO2 IP, one 10 cm dish of each cell line was lysed in 1 ml modRIPA buffer (10 mM Tris-cl pH 7.0, 150 mM NaCl, 1 mM EDTA, 1% Triton X-100, and 0.1% SDS) supplemented with proteinase inhibitors cocktail (Roche, 11873580001). Cell lysate was incubated with 50 μl SureBeads Protein G Magnetic Beads (Bio-Rad) plus 5 μg mouse anti-AGO1(2A7) (Wako, 015-22411) or anti-AGO2(4G8) (Wako, 015-22031) monoclonal antibody at 4 °C overnight with rotation. After washing 5 times with TBS buffer (50 mM Tris-HCl [pH 7.4], 150 mM NaCl) at room temperature, the beads were lysed in 1 ml Trizol (Life Technologies,15596-018) for RNA extraction. For ectopic AGO1 and AGO2 IP, cells were lysed 48 h post-transfection and followed the same protocol except using anti-Flag M2 beads (Sigma, M8823) instead.

## Western blotting

Cells were lysed in modRIPA buffer with protease inhibitor cocktail (Roche, 11873580001). The cell lysates were quantitated using a BCA kit (Pierce, 23225) in the Glomax multi+ machine. 30 μg protein of whole cell lysate or variable volume of Flag-IP samples was loaded into 4-20% Mini-PROTEAN® TGX Stain-Free™ Gels (Bio-Rad, 4568083) and then transferred onto a PVDF membrane using the Trans-Blot Turbo Transfer System. Primary antibodies used in this study are rabbit anti-ZCCHC11 (TUT4, Proteintech, 18980-1-AP, 1:500), rabbit anti-ZCCHC6 (TUT7, Proteintech, 25196-1-AP, 1:2000), rabbit anti-PAPD4 (TENT2, Abcam, AB103884, 1:500), mouse anti-Flag (Sigma, F1804, 1:3000) and mouse anti-α-tubulin (Sigma, T9026, 1:5000). The signals were developed with Pierce ECL plus Western Blotting Substrate (Pierce, 34080) and imaged by the Chemidoc Touch Imaging System.

## IP for ectopically expressed TUT4 and TUT7 proteins

pIRESneo-FLAG/HA-TUT4 or pIRESneo-FLAG/HA-TUT7 were transfected into HEK293T cells and cells were harvested in lysis buffer (20 mM Tris-HCl at pH 8.0, 137 mM NaCl, 1 mM EDTA, 1% Triton x-100, 10% Glycerol, 1.5 mM MgCl2, 5 mM DTT) 48 h post-transfection. Protein was purified using anti-Flag M2 beads (Sigma, M8823) and resuspended in Buffer D (200 mM KCl, 10 mM Tris-HCl [pH 8.0], 0.2 mM EDTA). The IP samples were confirmed by Western Blot analysis with anti-TUT4 (Proteintech, 18980-1-AP, 1:500), anti-TUT7 (Proteintech, 25196-1-AP, 1:2000) and anti-Flag antibody (Sigma, F1804, 1:3000).

## In vitro uridylation

miRNAs were synthesized by IDT and radio-labeled at the 5′-end with T4 polynucleotide kinase (NEB, M0201S) and ($γ$-$^{32}$P) ATP (Perkin Elmer, BLU502Z001MC) at 37 °C for 30 min. $^{32}$P-labeled miRNAs were purified by G25 columns (FISHER, 27-5325-01). In vitro uridylation reaction was performed in a total volume of 30 μl in 3.2 mM MgCl2, 1 mM DTT, 6 nM UTP, ~0.6 nM 5′-end labeled miRNA (~1 × 10$^4$ cpm), 1.33 U/μl RNase inhibitor (NEB, M0314L) and 15 μl of immunopurified proteins on beads in Buffer D. The reaction mixture was incubated at 37 °C for 15 min. 10 μl out of 30 μl of the RNA was denatured and run on a 20% urea polyacrylamide gel. The sequences of miRNAs are listed in Supplementary Table 2.

## Small RNA-seq library preparation

Small RNA libraries were generated using QIAseq miRNA Library Kit (QIAGEN, 331505) according to the manufacturer instructions, except that the library DNAs were size-selected and purified by running in a native 6% (w/v) acrylamide gel and followed by ethanol precipitation. Library quality was assessed using Qubit dsDNA HS Assay Kit (ThermoFisher, Q32854) and Agilent High Sensitivity DNA kit (Agilent, 5067-4626). Libraries were pooled together and sequenced on Illumina miSeq or NEXTseq platforms according to the manufacturer's specifications.

## MicroRNA expression and 3′ isomiR analysis

MiRNA expression levels and 3′ isomiR composition were analyzed using QuagmiR on the NCI Cancer Genomics Cloud[62]. For alignment, mismatches at 5′ and 3′-ends but not in the middle of reads are allowed.

For those miRNAs that present multiple paralogs copies, all genomic loci were considered. More details of the R analysis are reported on GitHub (https://github.com/Gu-Lab-RBL-NCI/TUT-tailing). A summary of miRNA expressions and isomiR composition can be found in Supplementary Data 1.

## Analysis of tail composition

Annotations from miRBase v22[63] were used to classify miRNA isoforms as canonical, trimmed (shorter than the annotated mature miRNA sequence), or having a tail (longer than the annotated mature miRNA sequence). In the latter case, the 3′ additional nucleotides are defined as tails. If the tail sequence does not match genomic sequence, the isomiR is classified as a NT isomiRs. Otherwise, it is categorized as an ambiguous isomiR. Moreover, NT tailing events were subdivided into three major categories: mono-adenylation, mono-uridylation, and mixed (containing all other mono-tailing and oligo-tailing possibilities). More details of the R analysis are reported on GitHub (https://github.com/Gu-Lab-RBL-NCI/TUT-tailing/tree/main/processing%20miRNA).

## Tailing cumulative curves

The global impact of knockout and rescue of TENT2/TUT4/TUT7 on tailing was measured using cumulative curves. Briefly, for each miRNA isoform a fold-change of abundance was calculated between the two experimental conditions. The resulting distribution of fold-changes obtained was sorted into bins representing the fraction of isomiRs with fold-changes equal to or smaller than the set threshold value for that bin. More details of the R analysis are reported on GitHub (https://github.com/Gu-Lab-RBL-NCI/TUT-tailing/tree/main/cumulative-curves-tailing).

## Statistics and reproducibility

The Wilcoxon test was used to determine $p$ values, and values <0.05 were considered statistically significant. Statistical analysis was done in RStudio (2022.07.1 Build 554) and GraphPad Prism v8 statistical software. Gel images in Western and Northern blots were selected from two or more representative experiments (Figs. 2a, d; 5f, g; 6d–g; Supplementary Figs. 3a; 5b; 6d,e). Code used during data analysis can be found at GitHub repository: https://github.com/Gu-Lab-RBL-NCI/TUT-tailing.

## Reporting summary

Further information on research design is available in the Nature Research Reporting Summary linked to this article.

# Data availability

The small RNA-seq data generated in this study have been deposited in the GEO database under accession code GSE183384, GSE184550, and GSE203472. The processed miRNA isomiR data generated in this study are provided in the Supplementary Information/Source Data file. Additional data that support the findings of this study are available from the corresponding author upon reasonable request. Source data are provided with this paper.

# Code availability

Code used during data analysis can be found at GitHub repository (https://github.com/Gu-Lab-RBL-NCI/TUT-tailing).

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

## Acknowledgements

This work has been funded by the intramural research program of the National Cancer Institute, National Institutes of Health (ZIA BC 011566) [A.Y., X.B.D.-R., R.S., TJ.S., P.V., S.G.]. The Seven Bridges Cancer Genomics Cloud has been funded in whole or in part with Federal funds from the National Cancer Institute, National Institutes of Health, Contract No. HHSN261201400008C and ID/IQ Agreement No. 17×146 under Contract No. HHSN261201500003I [X.B.D.-R, A.Y.].

## Author contributions

A.Y., X.B.-D.R., and S.G. designed the research. A.Y. performed all the experiments with help from R. S., TJ. S., and P.V.; X.B.-D.R. did all bioinformatic analyses; A.Y., X.B.-D.R., and S.G. analyzed the data. A.Y., X.B.-D.R., and S.G. wrote the paper.

## Funding

## Competing interests

The authors declare no competing interests.
