## [Peer Review File · Nature Communications]

Title: TENT2, TUT4, and TUT7 selectively regulate miRNA sequence and abundanceREVIEWER COMMENTS

Reviewer #1 (Remarks to the Author):

In this manuscript, Yang et al., use CRISPR KO of TENT2, TUT4 and TUT7 and miRNA sequencing to examine the 3' tailing specificity of each TENT and their effect on miRNA levels. With rigorous genetic tools in HEK293T cells, including individual and combinatorial KO of TENTs as well as rescue, the authors provided a comprehensive body of evidence to delineate the function of TENT2 in guanylation and uridylation, the more dominant role of TUT4 in uridylation than TUT7. They also identified several additional mir, such as mir181b and mir222, whose abundance is negatively regulated by TUT4/7 mediated uridylation, and presented evidence that mir888 is upregulated (stabilized) by TUT7. However, the current manuscript doesn't sufficiently advance the field by demonstrating the functional impact of these TENTs, either individually or combinatorially, on cellular activities or miRNA functions. For example, how do TENT KO cells replicate, survive or die? Does the change of several of mir lead to discernible changes in their target expression and function?

I have the following specific comments:

1. The authors used Ago2 IP followed by deep sequencing to characterize 3' tailing. Is it possible that different Ago may cause different 3' tailing behavior? Or 3' tailed mir may be released from Ago RISC. It'll be helpful if the authors can measure 3' tailing in either total mir pool or from Ago1/3 IP, and compare the results to Ago2 IPed mir.
2. It's a great effort by the authors to generate multiple single and combinatorial KO of the 3 TENTs. It'll be useful to examine whether these KO cells display any defects and whether their loss causes any measurable changed in the transcriptome. It'll be important to provide new insights into the key question – whether 3' tailing of mir have significant impact on cellular functions.
3. In Fig. 3, the authors showed that TENT2 KO by itself didn't affect uridylation but TENT2 KO in TUT4/7 KO background appears to further reduce uridylation. And this reduced uridylation can't be rescued by TENT2 overexpression. Were these experiments performed in Ago2 IPed mir pool? If yes, the authors should examine the tailing pattern of total mir pool.
4. In Fig. 2b and 2d, statistical analysis should be performed and the quantification of each mir signals should be quantified. In Fig. 4a, TENT2's rescue effect should also be statistically determined. In Fig. 5f-g, the level of each mir should be quantified as done for fig. 6.

Reviewer #2 (Remarks to the Author):

The manuscript by Yang et al entitled "TENT2, TUT4, and TUT7 selectively regulate miRNA sequence and abundance" comprehensively analyzes the effects of TENT2, TUT4, and TUT7 individual or combined knockouts in miRNAs in HEK293 cells. The results based largely on deep sequencing are reliable and confirm that the above-mentioned poly(A) and poly(U) polymerases can add untemplated nucleotides to the 3' end of miRNAs. Depending on the enzyme this can or not affect the miRNA abundance. The authors also indicate that in addition to A TENT2 can also decorate miRNA with G. Deep sequencing is also supplemented by northern blotting and in vitro activity assays. Rescue experiments further increase the reliability of the presented results.

The Authors are trying to emphasize the specificity of individual enzymes, which is apparent for the results, but unfortunately, the mechanism is not provided. Moreover, the biological effect of the regulation also remains to be established. Therefore the impact of this publication although it provides highly reliable data is somehow limited.

Minor points:

In many instances, the Authors use the expression depletion for the KO which is misleading.

The paper is difficult to follow as it catalogs the effects of KOs on miRNAs rather than performing synthesis.

Reviewer #3 (Remarks to the Author):

In their study "TENT2, TUT4, and TUT7 selectively regulate miRNA sequence and abundance", Yang and colleagues describe individual or combinational effects of TENT2, TUT4, and TUT7 enzymes in responsible for canonical miRNA tailing. They generated sets of isogenic single, double and triple knock out cells along with each individual rescue lines to explore selective nature of TENT mediated miRNA tailing. With these valuable sources, they performed extensive computational analysis, high quality biochemical assays and successfully identified different preference on miRNA 3' end tailing for these novel miRNA tailing enzymes. Surprisingly, they also found TUT4 and TUT7 have some distinct non-redundant effects. Overall, the selectivity of miRNA modifications across different genetic combinations is well documented, and there are likely a variety of mechanisms by which tailing can affect miRNA levels.

As a descriptive study, the limitation is on knowing the molecular mechanism for some of the miRNA changes. However, I think this work should be of interest to small RNA community, because of the large data resource provided and also the mutant cell lines generated. There are many labs studying miRNA tailing in mammals but there are some mixed conclusions from the field. Partly it is that tailing has different functional effects depending on the miRNA, but also some groups in the field used imperfect strategies (like knockdowns). In fact, it was suspected by some leading labs in the field that it was not possible to make such a triple mutant cell for lethality reasons.

This study reports that overall there isn't a fixed correlation between tailing and small RNA abundance.

But their use of genetic rescue analysis including wildtype and catalytic inactive variants makes the findings of greater value than many previous studies. The mutant cells will be very valuable foundation for the field, and they generate very extensive datasets. Therefore I think this work is a good candidate for publication on Nature Communications. I have some suggestions that may improve their conclusions and especially their data presentation since that is the major feature of the study.

Major comments.

- The most interesting aspect for the field is the data resource, but unless I missed it, the only two Supplementary Tables showed some qualitative summaries like positive or negative regulation by TUT4/7, and oligo sequences. And all of their heatmaps are showing only fold or relative change, but its not obvious the actual number of reads for each locus, which could vary a lot. It is necessary to provide sufficient supplementary data to support the tables and heatmaps that they could be redrawn, and explored further by interested readers. This can organize by the miRNA locus or isomiR, and contain information on types of untemplated 3' nucleotides and the changes in mutants. It could be kind of complicated tables, so I don't know how they organized it to generate all the heatmaps and scatterplots, but this is the important core of the paper results and to share with the field if published.
- Several main figures (Fig 3, Fig 5) have large heatmaps with sequences written alongside. These are difficult to read and in the print version won't be possible. The authors should put these in supplement and condense the main figures to focus on overall trends.

At the same time, it is very important to clearly provide the full data in the supplement and in table forms. As there were some questions raised by inspecting them. For example, sometimes we can see the same miRNA listed a few times in a heatmap. I assume its because a different isomiR is present, but somehow i should be more clear. Also in this example, supple figure 3d, multiple hsa-miR-128-3p-1-2 isoforms are plotted as below, and 5'end of this miRNA should be T rather than A. They didn't describe their mapping strategy in the method section, but it seems they allowed some mismatches. It would be better to explain about the details upfront. If there are many internal mismatch reads in the dataset, that could be a question if they are necessarily untemplated when at terminal.

5'-AA (left panel)

```
TCACAGTGAACCGGTCTCTTTAA
TCACAGTGAACCGGTCTCTTAA
ACACAGTGAACCGGTCTCTTTAA
```

5'-UU (middle panel)

```
ACACAGTGAACCGGTCTCTTTTT
```

5'-AU/UA (right panel)

```
TCACAGTGAACCGGTCTCTTTAT
```

TCACAGTGAACCGGTCTTTAT

- The categories of their analysis are a little difficult to follow. For example, it is not clear what is "templated tailing" and "templated isomiRs". Do they mean "templated" sequences are species that align to genome but actually modified by TENT enzymes? Maybe they need a different name than "templated" because I assume TENT enzymes always acting in an untemplated fashion? I think they are suggesting this in Figure 1E examples, but it was not clear how they distinguish it from alternative cleavage. This seems like a complex annotation problem that would need an external reference of Drosha or Dicer cleavage sites. So they should show more systematically how they determined sites of "templated" additions, with more example data and the re-annotated miRNA species.

Interestingly, it seems they found (for example Figure 3C) a shift in the genetic behavior of "templated" A and/or U when removing TENT2 from wt or TUT4/7 cells, which is really interesting. Especially, the contribution of TENT2 to U-tailed is greater in DKO than in wt cells (if I understood the graph). They should show some examples of the loci to show these behaviors, as it seems striking.

- Fig 5 compares the CDF of single TUT4 and TUT7 KO. I might miss it, but this figure should include CDF of the dKO to see if there is a larger effect, or is epistatic. For example, Fig 5C seems indicating miR-126-5p is the same in single KO and DKO, but miR-148b-3p, dKO is worse than either single KO TUT4 or TUT7. Also their conclusion is that TUT4 is doing most uridylation (not TUT7), but this is slightly controversial to their data that even TENT2 has substantial effects of uridylation on miRNAs that can only be detected in TUT4/7 mutant (Figure 3). Would it be safer to say there is a hierarchical substrate preference? I think it would be better if more comparisons, such as their bar graphs (Fig 5C), are showing the triple knockout.

- There are many correlation dot plots, but they are often hard to read without any reference. In Figure 4E-G, some miRNAs are labeled but the font is too small to read easily. In general, since there are a number of miRNAs that have previously studied or are studied in this manuscript as tailing targets, it makes sense to label so they can be a reference to interpret the figures. For example, in Fig 5C, some individual miRNAs behavior are shown in bar plots, but we don't know where they live in scatterplots.

These are highlighting why its hard for the reader to explore these data without providing the supplementary tables to document all of the analyses.

- I think other labs have observed that in TUT4/7 DKO there is an increase in A-tailing, possibly by TENT2 which has some overlap substrates also as shown here. It seems they didn't observe this (Figure 2B) or is it happening in some restricted set of miRNAs?

- In supple figure 3b, they have a large block of mono-U tailed miRNAs are listed on the mono-G category heatmap.

- According to the supple figure 3b and line 188-190 from the main text, they concluded that TENT2 has responsible for miRNA mono adenylation from miRNA specific manner, by showing TENT2-rescue patterns in both TENT2-KO and/or triple-KO cells, however the miRNA sets they used for rescue experiments (supple figure 3b) and knock out experiment (main figure 3b) seem different. For example, the highest rescued mono-A-tailed miRNAs, miR-30a-5p, miR-30d-5p in the supple figure 3b are not listed in the main figure 3b.

Similarly, many of U-mono tailed miRNAs rescued by TUT4,7,4/7 expression in supple figure 5e are not listed in the main figure 5a heatmap, as similar as above.

The detailed responses are outlined below. The reviewer's comments are in italic font and our responses are in blue. The manuscript was modified accordingly (in red).

Point-by-point response:

Reviewer #1 (Remarks to the Author):

In this manuscript, Yang et al., use CRISPR KO of TENT2, TUT4 and TUT7 and miRNA sequencing to examine the 3' tailing specificity of each TENT and their effect on miRNA levels. With rigorous genetic tools in HEK293T cells, including individual and combinatorial KO of TENTs as well as rescue, the authors provided a comprehensive body of evidence to delineate the function of TENT2 in guanylation and uridylation, the more dominant role of TUT4 in uridylation than TUT7. They also identified several additional mir, such as mir181b and mir222, whose abundance is negatively regulated by TUT4/7 mediated uridylation, and presented evidence that mir888 is upregulated (stabilized) by TUT7. However, the current manuscript doesn't sufficiently advance the field by demonstrating the functional impact of these TENTs, either individually or combinatorially, on cellular activities or miRNA functions. For example, how do TENT KO cells replicate, survive or die? Does the change of several of mir lead to discernible changes in their target expression and function?

Thanks for the positive comments! We have measured the impact of TENTs on cell proliferation as suggested. Please see answer to specific point #2 below for detail.

I have the following specific comments:

1. The authors used Ago2 IP followed by deep sequencing to characterize 3' tailing. Is it possible that different Ago may cause different 3' tailing behavior? Or 3' tailed mir may be released from Ago RISC. It'll be helpful if the authors can measure 3' tailing in either total mir pool or from Ago1/3 IP, and compare the results to Ago2 IPed mir.

This is a great point! As suggested, we sequenced small RNAs from HEK293T cells total RNAs and compared the result to that from the AGO2-IP. While miRNAs were enriched in AGO2-associated small RNAs as expected (Supplementary Fig. 1d), relative abundance of isomiRs did not change (Fig. 1f). Examination of top 100 expressed mono-U-tailed and mono-A-tailed isomiRs showed no consistent alteration between total miRNAs and miRNAs associated with the AGO2 (Supplementary Fig. 1e). While these results cannot rule out the possibility that there is a pool of AGO-free miRNAs, these miRNAs, if exist, are likely having a similar profile of 3' tailing as AGO2-associated miRNAs.

We also sequenced AGO1-associated miRNAs by IP of endogenous AGO1. No preference of uridylation or adenylation was observed between AGO1- and AGO2-associated miRNAs (Supplementary Fig. 1f). To ensure this is not due to a lack of specificity between anti-AGO1 and anti-AGO2 antibodies, we ectopically expressed Flag-tagged AGO1 and AGO2 in HEK293T cells separately and isolated associated sRNAs by FLAG-IP. Again, the isomiR profiles are comparable

between AGO1- and AGO2-associated miRNAs (Fig. 1g). These results indicate that the miRNA 3' modification machinery targets different AGOs analogously.

In addition, we observed an unexpected while intriguing result: Uridylated isomiRs were depleted in miRNAs associated with FLAG-AGOs compared to miRNAs associated with endogenous AGOs. This is consistent between AGO1 and AGO2 while being specific to uridylated but not adenylated isomiRs (Supplementary Fig. 1f). Given that newly produced miRNAs are enriched in transiently expressed FLAG-AGOs compared to endogenous AGOs, this result suggests that aged miRNAs are more uridylated than nascent miRNAs. It is possible that cells use uridylation but not adenylation as a way to distinguish miRNAs at different stages of their metabolism, for example, separating miRNAs which have already been involved in target-repression from “naive” ones.

We have included these results in the text, figures and supplementary figures in the revised manuscript. Relevant results are attached below for your convenience.

Figures 1f and 1g

Supplementary Figure 1d and 1e

Supplementary Figure 1f

2. It's a great effort by the authors to generate multiple single and combinatorial KO of the 3 TENTs. It'll be useful to examine whether these KO cells display any defects and whether their loss causes any measurable

changed in the transcriptome. It'll be important to provide new insights into the key question – whether 3' tailing of mir have significant impact on cellular functions.

This is another great suggestion. We agree that it is important to investigate the biological function of these TENTs and demonstrate their impacts on cellular functions. To this end, we examined the impact of their knockouts on cell proliferation. Specifically, we measured the time which it takes for cell number to double before reaching confluency. For each KO line, 8 replicates consisting of two independent colonies were tested. We found that cells grew at a slower rate compared to the WT cells when TENTs were lost (Supplementary Fig. 7a). To rule out the possibility that the reduced cell growth was due to off-target effects of genome editing, we performed rescue assay with either TENT2, TUT4 or TUT7 in TKO cells. Compared to expression of the corresponding catalytic-dead mutants, expression of TUT4 or TUT7 increased cell growth rate, whereas expression of TENT2 had opposite, if any, effect on cell proliferation (Supplementary Fig. 7b). Together, these results indicate that uridylation mediated by TUT4 and TUT7 promotes cell growth of HEK293T cells. This is consistent with previous studies performed in multiple cancer cell lines (PMID: 22118463, 24056962, 32442398,34949722), suggesting a general role of TUT4/7 in cell proliferation. These results were attached below for your convenience.

Supplementary Figure 7a and 7b

Additional studies are required to illustrate the role that miRNAs play in this process. We are actively investigating this by performing RNA-seq and CLIP in this set of isogenic cells. We believe that it would be best to thoroughly investigate this question in a separate study. We hope that this reviewer will agree with our decision.

3. In Fig. 3, the authors showed that TENT2 KO by itself didn't affect uridylation but TENT2 KO in TUT4/7 KO background appears to further reduce uridylation. And this reduced uridylation can't be rescued by TENT2 overexpression. Were these experiments performed in Ago2 IPed mir pool? If yes, the authors should examine the tailing pattern of total mir pool.

Thanks for the suggestions. We performed the suggested experiments and found that TENT2-mediated uridylation can be observed in both scenarios: comparing either total miRNAs or AGO2IP miRNAs between DKO and TKO cells. This suggests that the TENT2-mediated uridylation is not limited to AGO2-associated miRNAs. The new result is included in the Supplementary Figure 3f.

4. In Fig. 2b and 2d, statistical analysis should be performed and the quantification of each mir signals should be quantified. In Fig. 4a, TENT2's rescue effect should also be statistically determined. In Fig. 5f-g, the level of each mir should be quantified as done for fig. 6.

Thanks for pointing these out. We have redrawn Fig. 2b to make the comparison between KOs more clear. For WT and each KO cell line, the average percentage of miRNAs with either U-tail or A-tail was calculated by considering the top 200 expressed isomiRs. Since we have two independent colonies for each KO, two average values were obtained and plotted in the panel. Regarding the statistical analysis, we felt that using N=400 (2x200) which results in extremely small p values would be misleading. In our opinion, the N should be 2, which makes it invalid to perform a statistical test. Instead, we opted to show the two values for each KO in black dots in the panel and state clearly how these values were calculated in the figure legends. We hope that this reviewer will agree with our decision.

For Fig. 2d, we have quantified the signals and plotted these in Supplementary Figure 2e.

For Fig. 4a, we agree that it would be great if we could perform a statistical test. However, only two values (miRNA read counts) from two deep sequencing results (one for each colony) were obtained. The N=2 makes the statistical test invalid.

For Fig. 5f-g, we performed quantification as suggested and plotted the results in Supplementary Figures 5n and 5o.

Reviewer #2 (Remarks to the Author):

The manuscript by Yang et al entitled "TENT2, TUT4, and TUT7 selectively regulate miRNA sequence and abundance" comprehensively analyzes the effects of TENT2, TUT4, and TUT7 individual or combined knockouts in miRNAs in HEK293 cells. The results based largely on deep sequencing are reliable and confirm that the above-mentioned poly(A) and poly(U) polymerases can add untemplated nucleotides to the 3' end of miRNAs. Depending on the enzyme this can or not affect the miRNA abundance. The authors also indicate that in addition to A TENT2 can also decorate miRNA with G. Deep sequencing is also supplanted by northern blotting and in vitro activity assays. Rescue experiments further increase the reliability of the presented results.

The Authors are trying to emphasize the specificity of individual enzymes, which is apparent for the results, but unfortunately, the mechanism is not provided. Moreover, the biological effect of the regulation also remains to be established. Therefore the impact of this publication although it provides highly reliable data is somehow limited.

We appreciate the reviewer's comment regarding the complexity of molecular mechanisms triggered by the addition of non-templated nucleotides to miRNAs. In addition to the work of many labs, our group has shown the impact of mono-uridylation on expanding the miRNA target repertoire (Yang et al. Mol Cell 2019), whereas oligo-uridylation can label mature miRNAs for DIS3L2-mediated

degradation (Yang et al. Nat. Comm. 2020). Although at that time we didn't explicitly distinguish between the roles of TUT4 and TUT7, we had insights suggesting some functional specificities. We felt it would be interesting and a useful resource for the field to do a thorough characterization on the common and different activities of the TENTs studied. Although it is not included in the current manuscript, we have ongoing work on the role of TENTs on targeting, biogenesis and decay of miRNAs.

We also appreciate the reviewer's concern about the biological effects of TENTs loss of function. To this end, we provide new evidence showing their effects on cell proliferation (Supplementary Fig. 7). We found that cells grew at a slower rate compared to the WT cells when TENTs were lost (Supplementary Fig. 7a). To rule out the possibility that the reduced cell growth was due to off-target effects of genome editing, we performed rescue assay with either TENT2, TUT4 or TUT7 in TKO cells. Compared to expression of the corresponding catalytic-dead mutants, expression of TUT4 or TUT7 increased cell growth rate, whereas expression of TENT2 had opposite, if any, effect on cell proliferation (Supplementary Fig. 7b). Together, these results indicate that uridylation mediated by TUT4 and TUT7 promotes cell growth of HEK293T cells. This is consistent with previous studies performed in multiple cancer cell lines (PMID: 22118463, 24056962, 32442398,34949722), suggesting a general role of TUT4/7 in cell proliferation. The results were attached below for your convenience.

Supplementary Fig. 7a and 7b

Minor points:

In many instances, the Authors use the expression depletion for the KO which is misleading.

Thanks for pointing this out. We apologize for not stating this clearly in the previous submission. We have revised the manuscript accordingly by replacing “depletion” with “knockout” or “loss”.

The paper is difficult to follow as it catalogs the effects of KOs on miRNAs rather than performing synthesis.

Thanks for the comment. We hope that our study, including the large data and isogenic KO cell lines generated, will be a valuable resource to the broader small RNA community. In this revision, we have included raw data for each panel/plot so that interested readers can redraw and explore these results in detail. We also modified multiple panels to improve clarity and readability.

Reviewer #3 (Remarks to the Author):

In their study “TENT2, TUT4, and TUT7 selectively regulate miRNA sequence and abundance”, Yang and colleagues describe individual or combinational effects of TENT2, TUT4, and TUT7 enzymes in responsible for canonical miRNA tailing. They generated sets of isogenic single, double and triple knock out cells along with each individual rescue lines to explore selective nature of TENT mediated miRNA tailing. With these valuable sources, they performed extensive computational analysis, high quality biochemical assays and successfully identified different preference on miRNA 3'end tailing for these novel miRNA tailing enzymes. Surprisingly, they also found TUT4 and TUT7 have some distinct non-redundant effects. Overall, the selectivity of miRNA modifications across different genetic combinations is well documented, and there are likely a variety of mechanisms by which tailing can affect miRNA levels.

As a descriptive study, the limitation is on knowing the molecular mechanism for some of the miRNA changes. However, I think this work should be of interest to small RNA community, because of the large data resource provided and also the mutant cell lines generated. There are many labs studying miRNA tailing in mammals but there are some mixed conclusions from the field. Partly it is that tailing has different functional effects depending on the miRNA, but also some groups in the field used imperfect strategies (like knockdowns). In fact, it was suspected by some leading labs in the field that it was not possible to make such a triple mutant cell for lethality reasons.

This study reports that overall there isn't a fixed correlation between tailing and small RNA abundance. But their use of genetic rescue analysis including wildtype and catalytic inactive variants makes the findings of greater value than many previous studies. The mutant cells will be a very valuable foundation for the field, and they generate very extensive datasets. Therefore I think this work is a good candidate for publication on Nature Communications. I have some suggestions that may improve their conclusions and especially their data presentation since that is the major feature of the study.

Really appreciate the positive comments!

Major comments.

- The most interesting aspect for the field is the data resource, but unless I missed it, the only two Supplementary Tables showed some qualitative summaries like positive or negative regulation by TUT4/7, and oligo sequences. And all of their heatmaps are showing only fold or relative change, but it's not obvious the actual number of reads for each locus, which could vary a lot. It is necessary to provide sufficient supplementary data to support the tables and heatmaps that they could be redrawn, and explored further by interested readers. This can organize by the miRNA locus or isomiR, and contain information on types of untemplated 3' nucleotides and the changes in mutants. It could be kind of complicated tables, so I don't know how they organized it to generate all the heatmaps and scatter plots, but this is the important core of the paper results and to share with the field if published.*

Thanks for the suggestion. We agree that one of the most interesting aspects of this study is the comprehensive set of NGS data. In addition to raw data and processed files which were deposited at GEO (access numbers listed in the data availability section), we included in this revision a supplementary table 3 as suggested. It contains detailed information of all isomiR analyzed in this study. For each isomiR, it provides sequence, read counts, trimm/tail length, tail sequence/composition etc, which will be of great value for interested readers to perform additional analyses. Furthermore, specific data sets used to generate each panel/plot were deposited in the Mendeley (access number listed in the *Data availability* section). Readers can redraw and explore further any heatmaps/scatter-plots in this manuscript.

- *Several main figures (Fig 3, Fig 5) have large heatmaps with sequences written alongside. These are difficult to read and in the print version won't be possible. The authors should put these in supplement and condense the main figures to focus on overall trends.*

We appreciate the suggestion. We replaced all heatmaps in main figures with a condensed version as suggested. The full version with sequence labels were moved to the supplementary figures for further detail.

At the same time, it is very important to clearly provide the full data in the supplement and in table forms. As there were some questions raised by inspecting them. For example, sometimes we can see the same miRNA listed a few times in a heatmap. I assume its because a different isomiR is present, but somehow it should be more clear. Also in this example, supple figure 3d, multiple hsa-miR-128-3p-1-2 isoforms are plotted as below, and 5'end of this miRNA should be T rather than A. They didn't describe their mapping strategy in the method section, but it seems they allowed some mismatches. It would be better to explain about the details upfront. If there are many internal mismatch reads in the dataset, that could be a question if they are necessarily untemplated when at terminal.

5'-AA (left panel)

TCACAGTGAACCGGTCTCTTTAA

TCACAGTGAACCGGTCTCTTAA ACACAGTGAACCGGTCTCTTTAA

5'-UU (middle panel) ACACAGTGAACCGGTCTCTTTTT

5'-AU/UA (right panel)

TCACAGTGAACCGGTCTCTTTAT

TCACAGTGAACCGGTCTCTTAT

Thanks for pointing this out. Indeed, our analysis pipeline, "QuagmiR", is a customized mapping algorithm for annotating isomiRs that allows different number of mismatches on the 5' and 3' end of miRNAs (Bofill-De Ros et al. *Bioinformatics* 2018). It is known that 5' modifications are rare, and most of them can be attributed to alternative cleavage events of Drosha and Dicer. To capture isomiRs resulting from Drosha/Dicer alternative cleavages, we set our pipeline to allow mismatches

at both 5' and 3' ends of the alignment, but no mismatches at the middle. We made it clear in the method in this revision.

Despite multiple filtering steps embedded in our “QuagmiR” pipeline to capture only bona fide isomiRs, as you pointed out, there are apparently still reads resulting from sequencing error or mismapping. To examine this carefully, we revisited the data which were used to generate all the heatmaps. Out of 1580 tailing events reported, we identified 17 isoforms (including 2 mir-128-3p isomiRs as you mentioned above) in which the mismatches on the 5' portion that can't be attributed to alternative cleavage events occurred during their biogenesis. These 17 reads are listed below. Intriguingly, 13 of them were a transition from T → A, most likely an artifact introduced during library cloning. While we cannot be certain of their origin, these are extremely rare events. Most importantly, removing these reads from our analyses does not change any of our conclusions. We therefore opted to remove these from our heatmaps. Nonetheless, the raw data can be found in the new supplementary figure 3 and be available for those who want to investigate this phenomenon in the future. The data tables used to generate new heatmaps were deposited at the Mendeley Data.

Internal editing	3'end tail
hsa-miR-148b-3p [ΔCAGTGCATCACAGAACTTTGTTT] ref. [TCAGTGCATCACAGAACTTTGT]	TT
hsa-miR-128-3p-1-2 [ΔCACAGTGAACCGGTCTCTTTT] ref. [TCACAGTGAACCGTCTCTTT]	TT
hsa-miR-26a-5p-1-2 [ΔTCAAGTAATCCAGGATAGGCTTT] ref. [TTC AAGTAATCCAGGATAGGCT]	TT
hsa-miR-26b-5p [ΔTCAAGTAATTCAGGATAGGTTT] ref. [TTC AAGTAATTCAGGATAGGT]	TT
hsa-miR-186-5p [ΔAAAGAATTCTCCTTTTGGGCTTT] ref. [CAAAGAATTCTCCTTTTGGGCT]	TT
hsa-miR-34a-5p [ΔGGCAGTGTCTTAGCTGGTTGTTT] ref. [TGGCAGTGTCTTAGCTGGTTGT]	TT
hsa-miR-629-5p [ΔGGGTTTACGTTGGGAGAACTTT] ref. [TGGGTTTACGTTGGGAGAACT]	TT
hsa-miR-769-5p [ΔGAGACCTCTGGGTTCTGAGCTTT] ref. [TGAGACCTCTGGGTTCTGAGCT]	TT
hsa-miR-7-5p-1-2-3 [ΔGGAAGACTAGTGATTTTGTGTTT] ref. [TGG AAGACTAGTGATTTTGTGT]	TT
hsa-miR-186-5p [CAAGGAATTCTCCTTTTGGGCTTT] ref. [CAAAGAATTCTCCTTTTGGGCT]	TT

hsa-miR-454-3p [AAGTGCAATATTGCTTATAGGGTT] ref. [TAGTGCAATATTGCTTATAGGGT]	TT
hsa-miR-92a-3p-1-2 [AATTGCACTTGTCCTCCGGCCTGTAA] ref. [TATTGCACTTGTCCTCCGGCCTGT]	AA
hsa-miR-128-3p-1-2 [ACACAGTGAACCGGTCTCTTAA] ref. [TCACAGTGAACCGGTCTCTTT]	AA
hsa-miR-4497 [GCTCCGGGACGGCTGGGAA] ref. [cCTCCGGGACGGCTGGGC]	AA
hsa-miR-196a-5p-1-2 [AAGGTAGTTTCATGTTGTTGGGAA] ref. [TAGGTAGTTTCATGTTGTTGGG]	AA
hsa-miR-196b-5p [AAGGTAGTTTCCTGTTGTTGGGAA] ref. [TAGGTAGTTTCCTGTTGTTGGG]	AA
hsa-miR-4497 [GGCTCCGGGACGGCTGGGAA] ref. [acCTCCGGGACGGCTGGGC]	AA

• *The categories of their analysis are a little difficult to follow. For example, it is not clear what is "templated tailing" and "templated isomiRs". Do they mean "templated" sequences are species that align to genome but actually modified by TENT enzymes? Maybe they need a different name than "templated" because I assume TENT enzymes always acting in an untemplated fashion? I think they are suggesting this in Figure 1E examples, but it was not clear how they distinguish it from alternative cleavage. This seems like a complex annotation problem that would need an external reference of Drosha or Dicer cleavage sites. So they should show more systematically how they determined sites of "templated" additions, with more example data and the re-annotated miRNA species.*

We apologize for the poor choice of words and the confusion it brought. In the original manuscript, we defined isomiRs which are longer than annotated (miRBase v22) as being "tailed" regardless of the origin of these additional nucleotides. We agree that "tailing" or "tailed" which was defined as an action of TENTs should not be used together with "templated" to avoid any confusion. In this revision, we defined these additional nucleotides as "tails". Depending on whether or not the tail sequence matches the genomic sequence, isomiRs can be categorized into isomiRs with nontemplated tails (referred as non-templated isomiRs) and isomiRs with templated tails. While isomiRs with non-templated tails are clearly a result of TENT-mediated modifications, the origin of isomiRs with "templated tails" is ambiguous. Some of these tails are added by TENTs in a nontemplated manner but nonetheless happen to match the genomic sequence. Others arise from alternative choices of cleavage sites by Drosha and Dicer during biogenesis. Hence, we referred to the latter as isomiRs with ambiguous tail or ambiguous isomiRs for short. We made this clear in the Methods section and in the main text by adding explanations in the Introduction. Figures were modified accordingly.

Interestingly, it seems they found (for example Figure 3C) a shift in the genetic behavior of "templated" A and/or U when removing TENT2 from wt or TUT4/7 cells, which is really interesting. Especially, the contribution of TENT2 to U-tailed is greater in DKO than in wt cells (if I understood the graph). They should show some examples of the loci to show these behaviors, as it seems striking.

We apologize for not stating this clearly in the original manuscript. These “templated” A/U isomiRs are isomiRs with ambiguous tails (ambiguous isomiRs). Since TENTs-mediated tailing is responsible for part of these ambiguous isomiRs, they followed the same pattern, albeit to a lesser extent, as non-templated isomiRs when various TENTs were knocked out.

In Fig. 3c, we found that knocking out TENT2 in WT HEK293T cells had no impact on the level of mono-uridylyated isomiRs (both non-templated and ambiguous). This is consistent with previous studies, and potentially one of reasons why TENT2-mediated uridylation on mature miRNAs has never been reported before. However, knocking out TENT2 on top of TUT4/7 knockouts led to a consistent and significant reduction ($p=5.5e-127$) of uridylyated isomiRs (comparing DKO with TKO). This suggests TENT2-mediated uridylation activity that is masked by TUT4 and TUT7 in WT cells and only becomes observable when TUT4 and TUT7 are absent. Given that TENT2 shows negligible uridylation activity *in vitro*, this effect observed in cells is unlikely to be mediated by TENT2 directly. It remains to be examined whether TENT2 mediates uridylation in WT cells or if it is only a compensation mechanism to maintain uridylation on miRNAs. Nonetheless, our results indicate that TENT2, in addition to TUT4/7, should be considered when investigating the function of uridylation on miRNAs. Examples of individual miRNAs can be found in the new Fig. 5C as well as in raw data deposited in Mendeley Data.

- *Fig 5 compares the CDF of single TUT4 and TUT7 KO. I might miss it, but this figure should include CDF of the dKO to see if there is a larger effect, or is epistatic. For example, Fig 5C seems indicating miR-126-5p is the same in single KO and DKO, but miR-148b-3p, dKO is worse than either single KO TUT4 or TUT7. Also their conclusion is that TUT4 is doing most uridylation (not TUT7), but this is slightly controversial to their data that even TENT2 has substantial effects of uridylation on miRNAs that can only be detected in TUT4/7 mutant (Figure 3). Would it be safer to say there is a hierarchical substrate preference? I think it would be better if more comparisons, such as their bar graphs (Fig 5C), are showing the triple knockout.*

Thanks for pointing these out. We apologize for not stating this clearly in the original manuscript. To compare uridylation mediated by TUT4 to that by TUT7, we used miR-126-5p as an example to show the general pattern of most mature miRNAs (to be precise, 85 out of 100 top expressed miRNAs with a non-templated mono-U-tail) that TUT4 uridylation is dominant. In contrast, miR-148b-3p was used as an example to show “exceptions” of this observation. For this group of miRNAs (15 out of top 100 expressed, Supplementary Table 1), TUT7 knockout had a bigger impact than TUT4 knockout. However, 14 out of 15 of these miRNAs are 3p-miRNAs, suggesting that TUT7 could be more robust in uridylyating pre-miRNAs. Regardless, when we measure all miRNAs as a whole, as indicated in Fig. 5b (upper two panels), the overall pattern is that TUT4 is more dominant than TUT7. These results suggest that TUT4 is responsible for uridylation of mature miRNAs whereas TUT7-mediated uridylation is limited to a subset of miRNAs and potentially

occurs only at the pre-miRNA stage. TENT2-mediated uridylation, as discussed in Figure 3, could not be observed when either TUT4 or TUT7 was present. Nonetheless, it became evident when TUT4 and TUT7 were absent.

To make these points clear, we added in this revision a CDF to compare the uridylation effect of TUT4-KO, TUT7-KO, DKO and TKO side-by-side (Supplementary Fig. 5h) to show that $TKO > DKO \sim TUT4-KO > TUT7-KO \sim$ baseline. We also added TKO value in Fig. 5c as suggested. In addition, we mentioned the “hierarchical substrate preference” idea in the discussion. The relevant figures were attached below.

Supplementary Fig. 5h

Fig. 5c

- There are many correlation dot plots, but they are often hard to read without any reference. In Figure 4EG, some miRNAs are labeled but the font is too small to read easily. In general, since there are a number of miRNAs that have previously studied or are studied in this manuscript as tailing targets, it makes sense to label so they can be a reference to interpret the figures. For example, in Fig 5C, some individual miRNAs behavior are shown in bar plots, but we don't know where they live in scatterplots.

These are highlighting why its hard for the reader to explore these data without providing the supplementary tables to document all of the analyses.

Thanks for the comments. We replaced the original labels with colored dots and moved the labels to figure legends. Nonetheless, we agree that it is challenging to interpret these rather complicated plots without having access to the raw data. In this revision, we deposited annotated raw data used to generate each plot in Mendeley Data. Readers can easily redraw these plots and locate any data points of interest.

- *I think other labs have observed that in TUT4/7 DKO there is an increase in A-tailing, possibly by TENT2 which has some overlap substrates also as shown here. It seems they didn't observe this (Figure 2B) or is it happening in some restricted set of miRNAs?*

Consistent with previous studies, we observed an increase in A-tailing when TUT4 and TUT7 were knocked out. To make it clear, we have redrawn Fig. 2b which is attached below. We did not perform statistical test due to the fact that there are only two data points (please see details in our answer to Reviewer 1 question #4). This effect can also be observed in Supplementary Fig. 5d (attached below). Interestingly, in both WT and TENT2-KO cells, knocking out TUT4/7 resulted in an increase in A-tailing, indicating that these A-tailings, at least in part, were mediated by TENTs in addition to TENT2. Nonetheless, since this effect was reported before and is subtle compared to other effects observed in our study, we opted to not explore it in detail other than mentioning it in the main text.

- *In supple figure 3b, they have a large block of mono-U tailed miRNAs are listed on the mono-G category heatmap.*

We sincerely apologize for this mistake. The large block of mono-U tailed miRNAs listed on the mono-G heatmap (Supplementary Figure 3b) was inherited from using the plot of mono-U tailed as a template. The error has now been corrected.

- According to the *supple figure 3b* and *line 188-190* from the main text, they concluded that *TENT2* has responsible for miRNA mono adenylation from miRNA specific manner, by showing *TENT2-rescue* patterns in both *TENT2-KO* and/or *triple-KO* cells, however the miRNA sets they used for rescue experiments (*supple figure 3b*) and knock out experiment (*main figure 3b*) seem different. For example, the highest rescued mono-A-tailed miRNAs, *miR-30a-5p*, *miR-30d-5p* in the *supple figure 3b* are not listed in the main *figure 3b*.

Similarly, many of U-mono tailed miRNAs rescued by *TUT4,7,4/7* expression in *supple figure 5e* are not listed in the main *figure 5a* heatmap, as similar as above.

We appreciate reviewer's comments regarding the analyses on the miRNA-specificity of the tailing. Our claim regarding the miRNA specificity of "A" is based on the analysis shown in *Figure 4c*. There we show that, rescue of *TENT2* on the *TENT2-KO* or the *TKO* background, has a strong correlation with the level of effects observed in their depletion. Similarly, the miRNA-specificity of "U" is shown in *Supplementary Fig. 5e*. In these plots, each dot is one miRNA, the effect of rescue was plotted against the effect of knockout.

The heatmaps were generated to show the trend of changes. To this end, we focused on top expressed isomiRs. To show knockout effect, we listed top expressed isomiRs in the WT cells. To show rescue effect, we listed top expressed isomiRs in the corresponding KO cells. IsomiR expression varies in different KO cell lines, resulting in the "uncorresponded" positions between miRNAs used in different heatmaps. This did not affect our conclusion since the side-by-side comparisons were shown in *Fig. 4c* and *Supplementary Fig. 5e* as mentioned above. To further illustrate this point, we redraw a few heatmaps in which the list of isomiRs were aligned. As shown below, the patterns/conclusions remain unchanged. Finally, with raw data used to generate these heatmaps available in this revision, readers should be able to compare any isomiRs of interest across all samples.

REVIEWERS' COMMENTS

Reviewer #1 (Remarks to the Author):

The revised manuscript is significantly improved at two main points. First, they performed Ago1 and Ago2 specific IP and examined whether there is Ago specific 3' tailing. They found no significant differences between Ago2-associated miRNAs and total miRNAs. The same hold true for Ago1-associated miRNAs. Therefore, 3' end tailing is likely independent of individual Ago proteins. Second, they performed cell proliferation analysis to detect a role of TUT4/7 in promoting cell proliferation. I have no major issues with the revised manuscript. But I have two comments. First, it was reported that Ago1, Ago2 and Ago3 associate with a similar pool of miRNAs in mouse and human cells (Wang et al., Genes & Dev 2013). This study further showed that Ago1 and Ago2 associated miRNAs have similar 3' tailing patterns. It'll be useful to determine whether Ago1 and Ago2 associated miRNA profiles are also similar in HEK293T cells, and discuss whether Ago proteins do or do not affect miRNA metabolism in an Ago-specific manner. Second, it is interesting that they observed pro-proliferation effect of TUT4/7. The rescue assays were nicely done. They should use BrdU/EdU labeling followed by flow cytometry analysis to determine whether a specific cell cycle stage is affected and/or apoptotic cells are observed upon the deletion of TUT4/7. This should provide more detailed information for the defective cell proliferation upon TUT4/7 depletion.

Reviewer #2 (Remarks to the Author):

The authors have addressed most of the reviewer's comments, and in my opinion, the manuscript is suitable for publication.

Reviewer #3 (Remarks to the Author):

The authors have made a good effort to respond to the points raised during the first round of review, by us and by the other referees.

- The presentation of the figures and the supplements is improved. For example, the main figures are streamlined, and improved CDF plots are shown in a stacked view that make it easier to compare changes in the different mutants. The labeling of some figures is improved.

- Although all of these enzymes have been studied with respect to miRNA profiling, the quality of the underlying data from clean knockout backgrounds, as well as their thorough analysis, helps to reveal new regulatory principles. For example, their combination knockouts show that TENT2 contributes to miRNA uridylation, even though it is not a uridylase in vitro and TUT4/7 are known as the dominant miRNA uridylating enzymes.

- It is good that they provide the raw data and specific data tables used to generate the figures, which will aid future analysis of these interesting libraries by the community. This will be an interesting resource for the miRNA field, and I am sure the data will continue to be used by a number of laboratories.

The detailed responses are outlined below. The reviewer's comments are in *italic* font and our responses are in blue. The manuscript was modified accordingly (in red).

Reviewer #1 (Remarks to the Author):

The revised manuscript is significantly improved at two main points. First, they performed Ago1 and Ago2 specific IP and examined whether there is Ago specific 3' tailing. They found no significant differences between Ago2-associated miRNAs and total miRNAs. The same hold true for Ago1-associated miRNAs. Therefore, 3' end tailing is likely independent of individual Ago proteins. Second, they performed cell proliferation analysis to detect a role of TUT4/7 in promoting cell proliferation. I have no major issues with the revised manuscript.

Thanks for the positive comments! We are glad to know that previous concerns were adequately addressed.

But I have two comments. First, it was reported that Ago1, Ago2 and Ago3 associate with a similar pool of miRNAs in mouse and human cells (Wang et al., Genes & Dev 2013). This study further showed that Ago1 and Ago2 associated miRNAs have similar 3' tailing patterns. It'll be useful to determine whether Ago1 and Ago2 associated miRNA profiles are also similar in HEK293T cells, and discuss whether Ago proteins do or do not affect miRNA metabolism in an Ago-specific manner.

As suggested, we compared small RNAs associated with AGO1 and AGO2 and find the miRNA profiles are similar. This result is included in the revision as supplementary figure 1f. We discuss this result and add suggested reference as following in the text:

“Consistent with previous studies^{39,40}, the overall miRNA expression profiles are also similar (Supplementary Fig. 1f), supporting the notion that miRNAs are not specifically sorted into different AGOs in mammalian cells.”

Second, it is interesting that they observed pro-proliferation effect of TUT4/7. The rescue assays were nicely done. They should use BrdU/EdU labeling followed by flow cytometry analysis to determine whether a specific cell cycle stage is affected and/or apoptotic cells are observed upon the deletion of TUT4/7. This should provide more detailed information for the defective cell proliferation upon TUT4/7 depletion.

This is a great point. It would be nice to know how exactly cell proliferation is regulated by TUT4/7. However, it is out of the scope of this study. Furthermore, the suggested cell cycle study on TENT knocking out cells by itself might be insufficient to answer this question due to concerns of off-targeting effect of genome editing. Rescue assays might be required. To this end, we need to establish stable expression of individual TENT in the triple KO background which may take extensive amount of effort and time. We feel that it would be better to address

this question in a separate, thorough study. Nonetheless, we appreciate this point and add the following in the text.

“Nonetheless, future studies are required to illustrate how exactly cell proliferation is affected and more importantly, the role that miRNAs play in this process.”

Reviewer #2 (Remarks to the Author):

The authors have addressed most of the reviewer's comments, and in my opinion, the manuscript is suitable for publication.

Thanks for your support!

Reviewer #3 (Remarks to the Author):

The authors have made a good effort to respond to the points raised during the first round of review, by us and by the other referees.

- The presentation of the figures and the supplements is improved. For example, the main figures are streamlined, and improved CDF plots are shown in a stacked view that make it easier to compare changes in the different mutants. The labeling of some figures is improved.
- Although all of these enzymes have been studied with respect to miRNA profiling, the quality of the underlying data from clean knockout backgrounds, as well as their thorough analysis, helps to reveal new regulatory principles. For example, their combination knockouts show that TENT2 contributes to miRNA uridylation, even though it is not a uridylase in vitro and TUT4/7 are known as the dominant miRNA uridylating enzymes.
- It is good that they provide the raw data and specific data tables used to generate the figures, which will aid future analysis of these interesting libraries by the community. This will be an interesting resource for the miRNA field, and I am sure the data will continue to be used by a number of laboratories.

Really appreciate the positive comments!